# The autophagy gene *Atg16l1* differentially regulates $T_{reg}$ and $T_H2$ cells to control intestinal inflammation

Agnieszka M Kabat[1], Oliver J Harrison[1,2‡], Thomas Riffelmacher[3,4‡], Amin E Moghaddam[1], Claire F Pearson[5], Adam Laing[6], Lucie Abeler-Dörner[6], Simon P Forman[7], Richard K Grencis[7], Quentin Sattentau[1], Anna Katharina Simon[3,4], Johanna Pott[1*†], Kevin J Maloy[1*†]

[1]Sir William Dunn School of Pathology, University of Oxford, Oxford, United Kingdom; [2]Immunity at Barrier Sites Initiative, National Institute of Allergy and Infectious Diseases, National Institutes of Health, Bethesda, United States; [3]MRC Human Immunology Unit, Weatherall Institute of Molecular Medicine, University of Oxford, Oxford, United Kingdom; [4]John Radcliffe Hospital, University of Oxford, Oxford, United Kingdom; [5]Kennedy Institute of Rheumatology, University of Oxford, Oxford, United Kingdom; [6]Peter Gorer Department of Immunobiology, King's College London, London, United Kingdom; [7]Faculty of Life Sciences, The University of Manchester, Manchester, United Kingdom

*For correspondence: johanna. pott@path.ox.ac.uk (JP); kevin. maloy@path.ox.ac.uk (KJM)

†These authors contributed equally to this work
‡These authors also contributed equally to this work

Competing interests: The authors declare that no competing interests exist.

**Abstract** A polymorphism in the autophagy gene *Atg16l1* is associated with susceptibility to inflammatory bowel disease (IBD); however, it remains unclear how autophagy contributes to intestinal immune homeostasis. Here, we demonstrate that autophagy is essential for maintenance of balanced $CD4^+$ T cell responses in the intestine. Selective deletion of *Atg16l1* in T cells in mice resulted in spontaneous intestinal inflammation that was characterized by aberrant type 2 responses to dietary and microbiota antigens, and by a loss of $Foxp3^+$ $T_{reg}$ cells. Specific ablation of *Atg16l1* in $Foxp3^+$ $T_{reg}$ cells in mice demonstrated that autophagy directly promotes their survival and metabolic adaptation in the intestine. Moreover, we also identify an unexpected role for autophagy in directly limiting mucosal $T_H2$ cell expansion. These findings provide new insights into the reciprocal control of distinct intestinal $T_H$ cell responses by autophagy, with important implications for understanding and treatment of chronic inflammatory disorders.

## Introduction

Crohn's disease (CD) and ulcerative colitis (UC) are the two most common forms of inflammatory bowel disease (IBD), characterized by chronic inflammation of the gastrointestinal tract. IBD is a complex multifactorial disease that emerges on a background of many genetic and environmental factors (*Maloy and Powrie, 2011*). In recent years, tremendous efforts have been undertaken to identify the genetic factors that influence susceptibility to IBD. In particular, genome-wide association studies (GWAS) and subsequent meta-analyses have identified over 150 distinct loci that influence IBD susceptibility, many of which have revealed novel pathways in disease pathogenesis (*Van Limbergen et al., 2014*). Among these, a single-nucleotide polymorphism (SNP) in the essential macroautophagy (hereafter called 'autophagy') gene *ATG16L1* was associated with an increased risk of CD (*Hampe et al., 2007*; *Rioux et al., 2007*). A recent study showed that the IBD predisposing T300A mutation in the coding region of *ATG16L1* led to increased degradation of ATG16L1 protein and reduced autophagy (*Murthy et al., 2014*), indicating that decreased autophagy may contribute to

**eLife digest** The gut presents a puzzle to our immune system. Immune cells must rapidly respond to antigens produced by harmful bacteria, but food and the beneficial bacteria that inhabit the gut also produce antigens that our immune system must tolerate. Inappropriate immune responses in the gut can lead to inflammatory bowel disease, a debilitating disease with no current cure. We do not fully understand why these harmful inflammatory responses arise, but we know that genetic factors are important. Mutations in genes that affect a process known as autophagy – a pathway that breaks down and recycles unwanted material inside cells – make inflammatory bowel disease more likely to develop, but exactly how they do so remains unclear.

T helper cells are crucial controllers of intestinal immune responses and changes in their numbers and behaviour occur during inflammatory bowel disease. Kabat *et al.* explored how the autophagy pathway affects these key immune cells in mice. Blocking autophagy in T cells altered the balance of different types of T helper cells in the gut. A crucial population of regulatory T cells, which keep inflammatory responses in check, was lost. At the same time, another population of T cells expanded: the T helper 2 ($T_H$2) cells that are responsible for driving allergies. As a result, the mice developed intestinal inflammation and produced antibodies against gut bacteria and food.

Overall, Kabat *et al.*'s results show that autophagy defects can alter the balance of different types of T cells in the gut, leading to inflammation in the intestine. These observations contribute to our understanding of how genetic changes may influence susceptibility to inflammatory bowel disease. They also suggest that drugs that activate autophagy could help to treat diseases associated with changes in regulatory T cells or $T_H$2 cells, including inflammatory bowel disease and allergies. It will now be important to test this and to confirm whether similar changes in T cells are present in humans that have mutations in autophagy genes.

IBD development. Polymorphisms in several other autophagy-related genes, including *IRGM, LRRK2* and *SMURF1*, are also linked to IBD susceptibility (*Van Limbergen et al., 2014*), suggesting that changes in the autophagy pathway alter intestinal homeostasis and predispose to chronic intestinal inflammation.

Autophagy is a highly conserved cellular process that targets cytoplasmic components for lysosomal degradation and maintains homeostasis by recycling damaged organelles and large cytoplasmic protein aggregates. Autophagy becomes particularly important during metabolic or infectious stress (*Mizushima, 2007*). Atg16l1 forms an essential autophagy complex with Atg5 and Atg12 that facilitates elongation of the initial isolation membrane that results in engulfment of the cargo and formation of the autophagosome. Subsequent fusion with the lysosome facilitates degradation and allows nutrient recycling (*Mizushima et al., 2003*). To identify the mechanisms through which autophagy may regulate intestinal tissue homeostasis, it is essential to understand the functional consequences of alterations in autophagy on both immune and tissue cells present in the gut. To date, several studies have examined the role of autophagy and Atg16l1 in intestinal epithelial cells and myeloid cells for intestinal homeostasis. In these studies, Atg16l1 was shown to play a role in Paneth cell physiology, as well as in bacterial handling and regulation of inflammatory IL-1β secretion by myeloid cells (*Cadwell et al., 2008; Kuballa et al., 2008; Saitoh et al., 2008; Plantinga et al., 2011*). However, the role of Atg16l1 in intestinal adaptive immune responses has not yet been addressed.

CD4[+] T cells constitute the largest population of intestinal lymphocytes and are central mediators of host protective and tolerogenic responses in the gut (*Shale et al., 2013*). In particular, thymus-derived and peripherally induced Foxp3[+] CD4[+] regulatory T cells (tT$_{reg}$ and pT$_{reg}$ cells, respectively) are indispensable in promoting tolerance toward commensal and dietary antigens and for the prevention of aberrant effector T cell responses, including $T_H$1, $T_H$2 and $T_H$17 cell responses (*Izcue et al., 2009*). An imbalance between effector and regulatory CD4[+] T cells can promote chronic intestinal inflammation and accumulation of effector CD4[+] T cells in the inflamed mucosa is a cardinal feature of IBD (*Abraham and Cho, 2009; Maloy and Powrie, 2011; Shale et al., 2013*).

Therefore, it is important to define factors that regulate aberrant CD4$^+$ T cell responses in the gastrointestinal tract.

Previous studies utilizing mice with T-cell-specific deletion of essential autophagy genes (*Atg3*, *Atg5*, *Atg7*, *Beclin1*) pointed to a key role of autophagy in T cell homeostasis, as these mice exhibited decreased frequencies and numbers of CD4$^+$ and CD8$^+$ T cells and defects in T cell proliferation in vitro (*Pua et al., 2009*; *Stephenson et al., 2009*; *Jia and He, 2011*; *Kovacs et al., 2012*). In addition, recent studies highlighted the importance of autophagy in the development of memory CD8$^+$ T cells (*Puleston et al., 2014*; *Xu et al., 2014*; *Schlie et al., 2015*). However, the exact requirements for autophagy during different stages of T cell activation and differentiation remain poorly understood (*Xu et al., 2014*). Given that the gastrointestinal tract is a site of continuous immune activation by external antigens and is therefore a challenging environment for the adaptive immune system, we hypothesized that a selective defect in autophagy may affect intestinal T cell homeostasis.

We investigated the role of *Atg16l1* in intestinal CD4$^+$ T cells by generating mice that selectively lack *Atg16l1* in T cells. Here, we show that T-cell-specific deletion of *Atg16l1* results in chronic intestinal inflammation accompanied by increased humoral responses toward commensal and dietary antigens. We further demonstrate that *Atg16l1*-deficiency has opposing effects on intestinal CD4$^+$ T cells subsets; markedly enhancing T$_H$2 responses whilst decreasing T$_{reg}$ cell numbers. Through selective ablation of *Atg16l1* in T$_{reg}$ cells, we established the importance of cell-intrinsic autophagy for intestinal T$_{reg}$ cell homeostasis. Furthermore, through complementary in vivo approaches we show that autophagy controls T$_H$2 responses through two distinct mechanisms; through a cell-intrinsic pathway and by promoting extrinsic regulation by T$_{reg}$ cells.

## Results

### Selective deletion of *Atg16l1* in T cells results in spontaneous intestinal pathology

To investigate the role of autophagy in intestinal T cell homoeostasis, mice carrying *lox*P-flanked alleles of the essential autophagy gene *Atg16l1* (*Atg16l1*$^{fl/fl}$) (*Hwang et al., 2012*) were crossed with *CD4-Cre* mice, generating *Atg16l1*$^{fl/fl}$::*CD4-Cre* mice (hereafter denoted as *Atg16l1*$^{\Delta CD4}$) in which *Atg16l1* is selectively ablated in T cells from the double-positive stage of thymic development. To verify functional deletion of *Atg16l1* autophagy levels were analyzed by autophagosome formation and LC3 lipidation. CD4$^+$ T cells isolated from control *Atg16l1*$^{fl/fl}$ mice exhibited increased LC3$^+$ autophagosome formation after activation, as measured by intracellular LC3 accumulation in the presence of a lysosomal inhibitor (*Figure 1A*). In contrast, there was no increase in intracellular LC3 accumulation in CD4$^+$ T cells from *Atg16l1*$^{\Delta CD4}$ mice (*Figure 1A*). To verify this finding using another method, we assessed LC3 lipidation by Western blot analysis (*Klionsky et al., 2012*). Activated control *Atg16l1*$^{fl/fl}$ CD4$^+$ T cells exhibited increased lipidated LC3 II levels in the presence of chloroquine, indicative of autophagy-mediated turnover of LC3 II after T cell activation (*Figure 1B*). However, LC3 II levels in CD4$^+$ T cells from *Atg16l1*$^{\Delta CD4}$ mice were barely affected by activation (*Figure 1B*), confirming a block in autophagy.

Young *Atg16l1*$^{\Delta CD4}$ mice appeared normal, initially gained weight in a manner comparable to *Atg16l1*$^{fl/fl}$ littermates and exhibited normal intestinal morphology (*Figure 1C,F-H*). However, from around 5 months of age, *Atg16l1*$^{\Delta CD4}$ mice stopped gaining weight (*Figure 1C*), developed splenomegaly and lymphadenopathy (*Figure 1D,E*) and chronic intestinal pathology that progressed with age (*Figure 1F–I*). *Atg16l1*$^{\Delta CD4}$ mice exhibited significant inflammation of both the small intestine (SI) and colon, characterized by increased SI length, marked lengthening of crypts, shortening of villi and epithelial hyperplasia (*Figure 1F–I*). Thus, T-cell-specific *Atg16l1* deletion resulted in spontaneous intestinal inflammation and systemic immune activation.

### *Atg16l1* deficiency has opposing effects on intestinal T$_{reg}$ and T$_H$2 cells

To characterize the effects of *Atg16l1* on intestinal and systemic T cell homeostasis independently from any confounding effects of ongoing tissue inflammation, we analyzed young (8–12 weeks old) *Atg16l1*$^{\Delta CD4}$ mice before the onset of inflammatory pathology or systemic symptoms. Whilst thymic T cell production was unperturbed in *Atg16l1*$^{\Delta CD4}$ mice (*Figure 2—figure supplement 1A,B*), frequencies of CD4$^+$ and CD8$^+$ T cells in peripheral lymphoid organs were significantly decreased

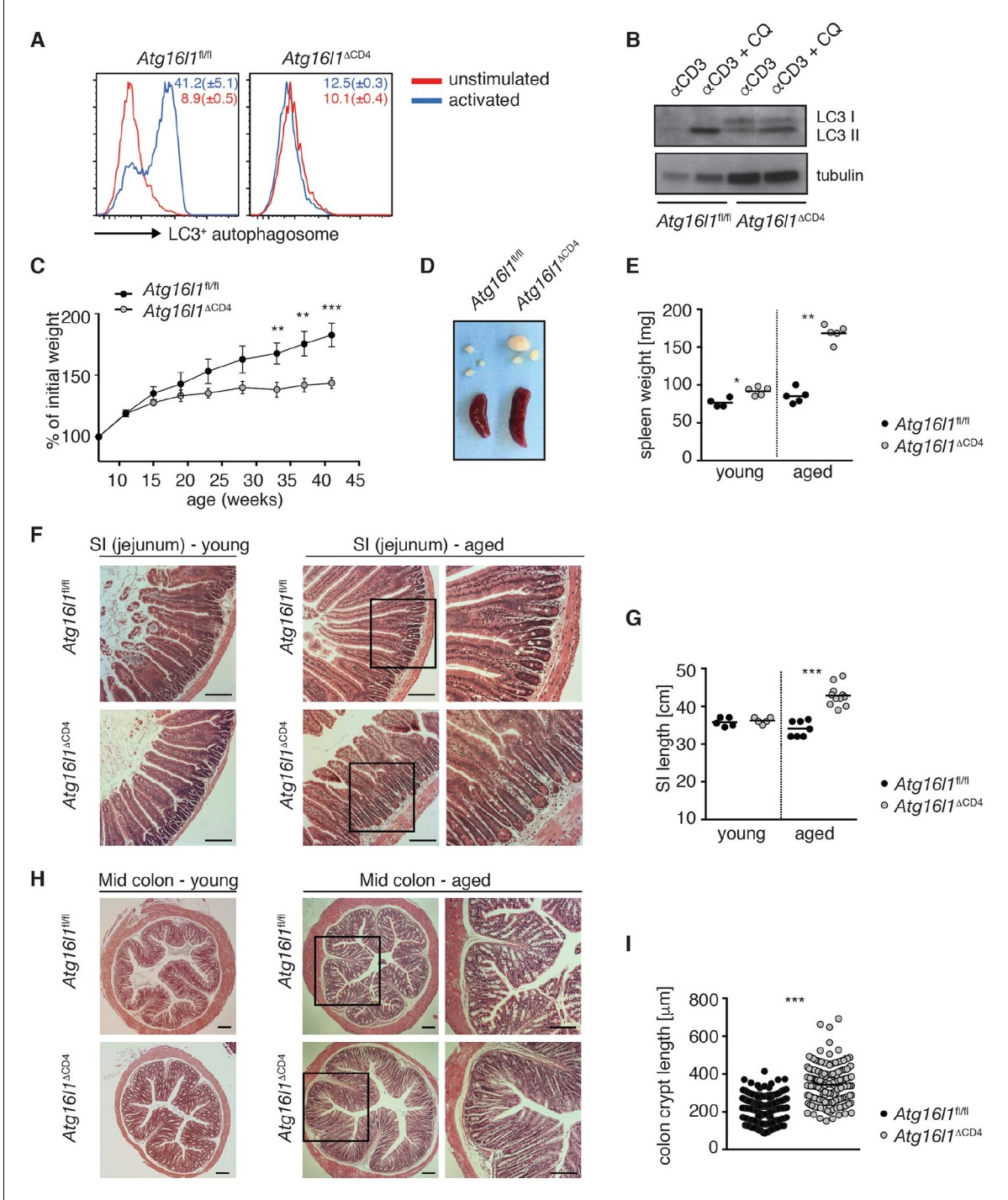

**Figure 1.** Aged $Atg16l1^{\Delta CD4}$ mice develop intestinal inflammation. (**A**) FACS analysis of LC3[+] autophagosome formation in CD4[+] T cells from cLP of $Atg16l1^{\Delta CD4}$ and $Atg16l1^{fl/fl}$ mice after overnight activation with or without α-CD3 (5 μg/ml) and α-CD28 (1 μg/ml). (**B**) Western blot analysis of LC3 lipidation in naïve splenic CD4[+] T cells isolated from $Atg16l1^{\Delta CD4}$ mice and $Atg16l1^{fl/fl}$ mice after 3hr activation with α-CD3 (5 μg/ml) and α-CD28 (1 μg/ml) with or without chloroquine (CQ, inhibitor of lysosomal degradation, 50 μM). (**C**) Weight curves of $Atg16l1^{\Delta CD4}$ and $Atg16l1^{fl/fl}$ littermates. (**D**) Representative images of spleens and mesenteric lymph nodes (mLN) from aged $Atg16l1^{\Delta CD4}$ and $Atg16l1^{fl/fl}$ littermates and (**E**) spleen weights of young and aged $Atg16l1^{\Delta CD4}$ and $Atg16l1^{fl/fl}$ littermates. (**F,H**) Representative photomicrographs of haemotoxilin and eosin (H&E) stained sections of (**F**) jejunum and (**H**) mid-colon from young and aged $Atg16l1^{\Delta CD4}$ and $Atg16l1^{fl/fl}$ littermates, scale bar 150 μm. (**G,I**) Quantification of (**G**) SI lengths and (**I**) mid-colon crypt lengths in aged $Atg16l1^{\Delta CD4}$ and $Atg16l1^{fl/fl}$ littermates. Data are representative of at least three independent experiments (**A-E, F, H**) or combined from two (**G**) or three (**I**) independent experiments, with at least 3 mice per group. Data shown as mean ± s.e.m (**A,C**). Each dot represents an individual mouse and horizontal bars denote means (**E,G**). In (**I**) each dot represents an individual crypt measurement and horizontal bars denote

*Figure 1 continued on next page*

*Figure 1 continued*

means. Statistical significance was determined using two-way analysis of variance (ANOVA) with Bonferroni's correction for multiple comparisons (**C**) or the Mann–Whitney test (**E,G,I**), \*\*p<0.01; \*\*\*p<0.001. SI LP– small intestine lamina propria, cLP – colonic lamina propria. Young mice: 8–12 weeks old, aged mice >5 months old.

compared to $Atg16l1^{fl/fl}$ littermates (*Figure 2A* and *Figure 2—figure supplement 1C*). Furthermore, we observed significant decreases in intestinal T cell frequencies and numbers in the cLP and SI LP of $Atg16l1^{\Delta CD4}$ mice (*Figure 2A* and *Figure 2—figure supplement 1D*). As $CD4^+$ T cells are the main drivers and regulators of chronic intestinal inflammation (*Shale et al., 2013*), we focused subsequent analyses on $CD4^+$ T cells.

Despite reduced numbers of T cells, $Atg16l1^{\Delta CD4}$ mice developed exacerbated disease in a $CD4^+$ T cell-mediated model of IBD, indicating that $Atg16l1$-deficient $CD4^+$ T cells are capable of driving intestinal inflammation (*Figure 2—figure supplement 2*). Analysis of the effector $CD4^+$ T cell compartment in $Atg16l1^{\Delta CD4}$ mice revealed that frequencies of colonic $T_H1$ ($IFN-\gamma^+$) and $T_H17$ ($IL-17A^{-+}$) populations were comparable in young $Atg16l1^{\Delta CD4}$ mice and $Atg16l1^{fl/fl}$ littermates (*Figure 2B, D*), although, due to decreased colonic $CD4^+$ T cell numbers, total $T_H1$ and $T_H17$ numbers were significantly decreased (*Figure 2C*). Conversely, both frequencies and total numbers of $T_H2$ ($IL-13^+$) cells were significantly increased in cLP of young $Atg16l1^{\Delta CD4}$ mice (*Figure 2B–D*). These IL-13-producing cells were *bona fide* $T_H2$ cells, as they co-expressed the lineage-specifying transcription factor Gata3 (*Figure 2D,E*). Interestingly, $T_H2$ cell accumulation was primarily observed in the intestinal mucosa of $Atg16l1^{\Delta CD4}$ mice, as $T_H2$ cell frequencies were only marginally increased in the mLN and remained unchanged in the spleen (*Figure 2E*). However, the functional consequences of $T_H2$ expansion extended beyond the intestine, as $Atg16l1^{\Delta CD4}$ mice had increased frequencies of eosinophils in both the spleen and mLN and elevated serum levels of mast cell protease 1 (MCPT-1), a marker of intestinal mast cell activation (*Figure 2—figure supplement 3A,B*).

As $Foxp3^+$ $T_{reg}$ cells play a non-redundant role in control of effector T cells and the development of intestinal inflammation (*Izcue et al., 2009*), we hypothesized that alterations in $T_{regs}$ might underlie the spontaneous intestinal pathology that developed in aged $Atg16l1^{\Delta CD4}$ mice. Indeed, we found that the frequencies of intestinal $Foxp3^+$ $T_{reg}$ cells in young $Atg16l1^{\Delta CD4}$ mice were severely reduced, both in the SI and cLP (*Figure 2F,G*). Taking into account the decreased frequencies of $CD4^+$ T cells in $Atg16l1^{\Delta CD4}$ mice (*Figure 2A*), this equated to a reduction in $T_{reg}$ cell numbers by around 10-fold in the colonic LP and 4-fold in SI LP (*Figure 2—figure supplement 4A*). In contrast, thymic development of $Foxp3^+$ $T_{reg}$ cells was not diminished in young $Atg16l1^{\Delta CD4}$ mice (*Figure 2—figure supplement 4B*), and we observed only minor, though significant, reductions in the frequencies and absolute numbers of $Foxp3^+$ $T_{reg}$ cells in the spleen and mLN of $Atg16l1^{\Delta CD4}$ mice compared with $Atg16l1^{fl/fl}$ littermates (*Figure 2G* and *Figure 2—figure supplement 4A*). Thus, $Atg16l1$-deficiency profoundly affected the maintenance of $Foxp3^+$ $T_{reg}$ cells in the periphery, particularly within the intestinal mucosa. Expression of neuropilin-1 (Nrp1) and Helios, putative markers proposed to distinguish $pT_{reg}$ and $tT_{reg}$ cells, were found at comparable levels on intestinal $Foxp3^+$ $T_{reg}$ cells from $Atg16l1^{fl/fl}$ and $Atg16l1^{\Delta CD4}$ mice, suggesting that the local environment, rather than site of $T_{reg}$ induction, primarily dictates the requirement for autophagy in $T_{reg}$ cells (*Figure 2—figure supplement 4C,D*). Assessment of how $Atg16l1$-deficiency affected intestinal $Foxp3^+$ $T_{reg}$ cell phenotype showed that impaired autophagy significantly increased expression of effector $T_H$ cytokines in $T_{reg}$ cells from cLP and SI LP (*Figure 2—figure supplement 4E*). We also found that cLP $T_{reg}$ cells from young $Atg16l1^{\Delta CD4}$ mice showed higher expression of CD103 and CTLA-4, but showed decreased expression of the activation markers CD25, CD69, and the terminal differentiation marker KLRG-1 (*Cheng et al., 2012*) (*Figure 2—figure supplement 4F*). In addition, intestinal $T_{reg}$ cells from young $Atg16l1^{\Delta CD4}$ mice had significantly increased expression of Ki67 and higher levels of phosphorylated S6, suggesting that the majority were in cell cycle (*Figure 2—figure supplement 4G–I*). Taken together, these results identify a crucial role for autophagy in the maintenance and functional regulation of intestinal $T_{reg}$ cells.

Overall, these results demonstrate that selective ablation of $Atg16l1$ in T cells led to a decrease in $Foxp3^+$ $T_{reg}$ cells and selective expansion of $T_H2$ cells that preceded the onset of overt pathology. In addition, these perturbations in $T_H$ cell subsets were largely limited to the mucosal environment.

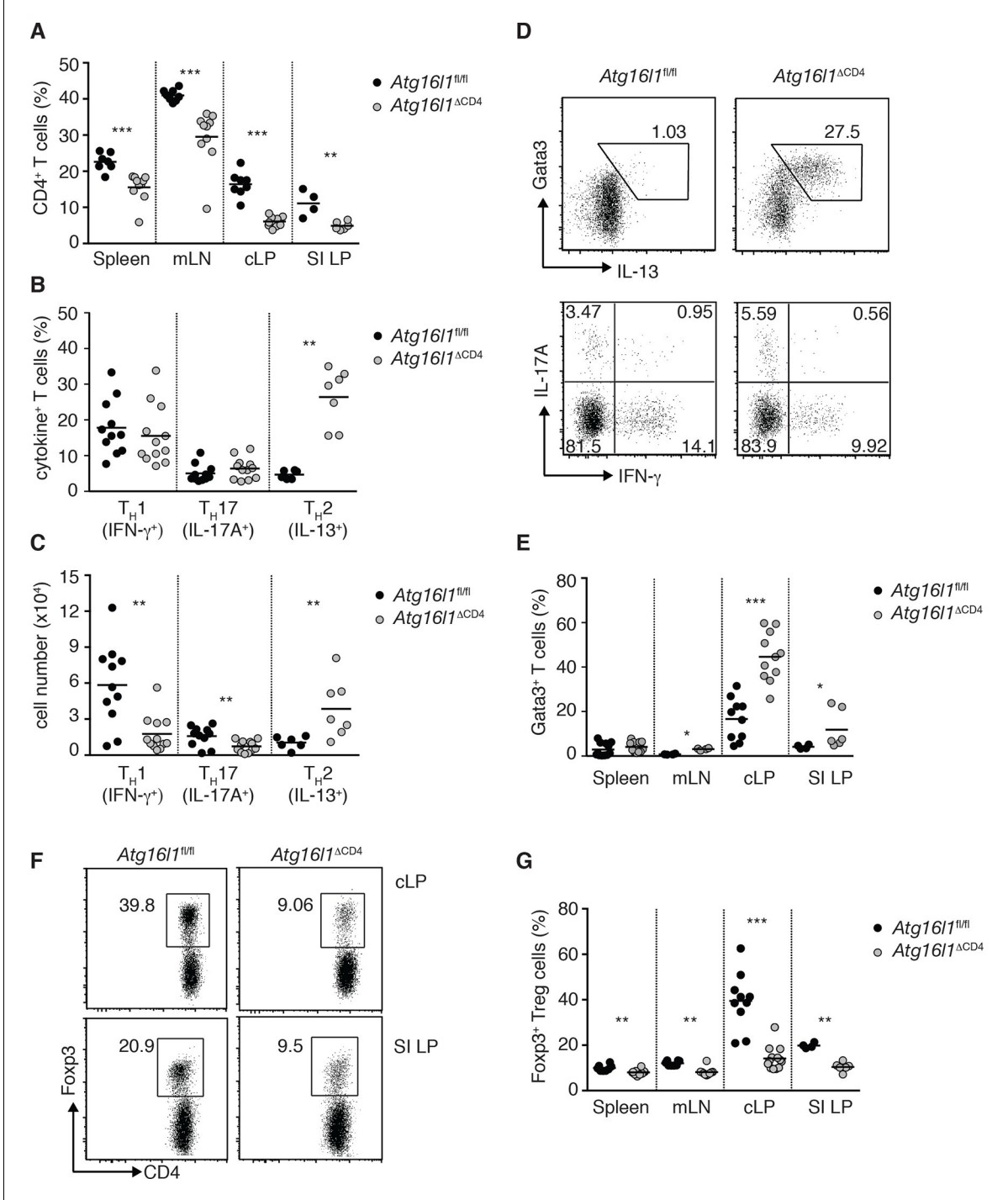

**Figure 2.** $Atg16l1^{\Delta CD4}$ mice exhibit reciprocal dysregulation of intestinal $T_H2$ and $T_{reg}$ cells before the onset of intestinal inflammation. (A) Frequencies of CD4+ T cells as a proportion of live cells in young $Atg16l1^{\Delta CD4}$ and $Atg16l1^{fl/fl}$ littermates. (B) Frequencies and (C) total numbers of IFN-γ+ $T_H1$, IL-17A+ $T_H17$ and IL-13+ $T_H2$ cells isolated from cLP of young $Atg16l1^{\Delta CD4}$ and $Atg16l1^{fl/fl}$ littermates (gated on CD4+ T cells). (D) Representative FACS plots of Gata3 and IL-13 (top) or IFN-γ and IL-17A (bottom) expression by cLP CD4+ T cells isolated from young $Atg16l1^{\Delta CD4}$ and $Atg16l1^{fl/fl}$ littermates (gated on CD4+ TCRβ+ Foxp3- live cells). (E) Frequencies of Gata3+ CD4+ T cells in young $Atg16l1^{\Delta CD4}$ and $Atg16l1^{fl/fl}$ littermates (gated on CD4+ TCRβ+ Foxp3- cells). (F) Representative FACS plots and (G) frequencies of Foxp3+ $T_{reg}$ cells in young $Atg16l1^{\Delta CD4}$ and $Atg16l1^{fl/fl}$ littermates (gated on CD4+ TCRβ+ cells). Data are combined from three or more independent experiments with at least two mice per group (A,B, D, E, G) or are representative of four independent experiments with at least four mice per group (D, F). Each dot represents an individual mouse and horizontal bars denote means. Numbers indicate percentage of cells in gates or quadrants. Statistical significance was determined using the Mann–Whitney test, *p<0.05; **p<0.01; ***p<0.001. SI LP– small intestine lamina propria, cLP – colonic lamina propria. Young mice: 8–12 weeks old.

*Figure 2 continued on next page*

*Figure 2 continued*

The following figure supplements are available for figure 2:

**Figure supplement 1.** Characterization of immune cell compartments in young $Atg16l1^{\Delta CD4}$ mice.

**Figure supplement 2.** $Atg16l1^{\Delta CD4}$ mice have increased susceptibility to T-cell-mediated experimental IBD.

**Figure supplement 3.** Elevated type 2 innate responses in $Atg16l1^{\Delta CD4}$ mice.

**Figure supplement 4.** Characterization of $Atg16l1$-deficient $T_{reg}$ cells.

## $Atg16l1^{\Delta CD4}$ mice exhibit elevated type 2 humoral responses to environmental antigens

We next assessed whether dysregulation in the intestinal $T_{reg}$ and $T_H2$ compartment in $Atg16l1^{\Delta CD4}$ mice affected humoral responses. While at the limit of detection in $Atg16l1^{fl/fl}$ controls, serum IgE concentrations were significantly elevated in young $Atg16l1^{\Delta CD4}$ mice and increased further as the mice aged (*Figure 3A*). Furthermore, levels of serum IgA and $IgG_1$ in young $Atg16l1^{\Delta CD4}$ mice were also significantly elevated relative to $Atg16l1^{fl/fl}$ littermates (*Figure 3—figure supplement 1A*) and again increased as the $Atg16l1^{\Delta CD4}$ mice aged (*Figure 3B*). In contrast, levels of isotypes not associated with $T_H2$ help were identical in aged $Atg16l1^{\Delta CD4}$ mice and $Atg16l1^{fl/fl}$ littermates (*Figure 3B*). Thus, there was a progressive dysregulation of $T_H2$-associated antibody responses in $Atg16l1^{\Delta CD4}$ mice. Consistent with these elevated humoral responses, young $Atg16l1^{\Delta CD4}$ mice had higher frequencies of germinal center (GC), memory B cells and plasma cells in the spleen and mLN compared to $Atg16l1^{fl/fl}$ littermates (*Figure 3—figure supplement 1B*) and markedly enlarged Peyer's patches were observed in aged $Atg16l1^{\Delta CD4}$ mice (*Figure 3C*).

Multiple studies have demonstrated the critical role played by $Foxp3^+$ $T_{reg}$ cells in immune tolerance to dietary and microbial antigens within the intestine. Furthermore, changes in intestinal $T_{reg}$ and $T_H2$ responses are associated with food hypersensitivities (*Berin and Sampson, 2013*). We hypothesized that the aberrant humoral responses in $Atg16l1^{\Delta CD4}$ mice might be directed against luminal antigens. Soy is the main protein source in chow, and we detected high levels of anti-soy $IgG_1$ and IgA in sera from aged $Atg16l1^{\Delta CD4}$ mice, whereas these responses were undetectable in control $Atg16l1^{fl/fl}$ littermates (*Figure 3D*). By contrast, we only detected marginal levels of soy-specific $IgG_{2b}$ or $IgG_{2c}$ in aged $Atg16l1^{\Delta CD4}$ sera (*Figure 3D*). Importantly, elevated anti-soy $IgG_1$ and IgA antibodies were already present in sera from young $Atg16l1^{\Delta CD4}$ mice, before the onset of intestinal inflammation (*Figure 3—figure supplement 1C*). Despite the very high levels of total serum IgE in aged $Atg16l1^{\Delta CD4}$ mice, we did not detect elevated levels of anti-soy IgE (data not shown). The absence of soy-specific IgE could be due to the inhibiting effects of persistent exposure to high-dose antigens on IgE responses (*Sudowe et al., 1997*; *Riedl et al., 2005*). Therefore, to test whether an IgE response was mounted during transient exposure to low-dose dietary antigens, we fed young $Atg16l1^{\Delta CD4}$ and $Atg16l1^{fl/fl}$ mice with ovalbumin (OVA), either alone or in combination with the mucosal adjuvant cholera toxin (CT). As expected, anti-OVA IgE responses were undetectable in control $Atg16l1^{fl/fl}$ mice fed OVA alone and were only marginally increased by co-administration of CT (*Figure 3E*). In contrast, $Atg16l1^{\Delta CD4}$ mice exhibited significantly elevated levels of anti-OVA IgE after being fed OVA alone and developed >10-fold higher levels of OVA-specific IgE after feeding of OVA with CT (*Figure 3E*). Together, these results indicate that $Atg16l1^{\Delta CD4}$ mice displayed aberrant $T_H2$-associated antibody responses towards otherwise innocuous dietary protein antigens.

Besides food antigens, the intestinal lumen harbors vast quantities of commensal-derived antigens. Thus, we measured antibodies directed against the flagellin antigen CBir1, produced by commensal bacteria belonging to *Clostridia* cluster XIVa, as antibodies against flagellin are readily detected in sera of IBD patients (*Lodes et al., 2004*). We detected significantly higher levels of CBir1-specific $IgG_1$ and IgA in the serum of aged $Atg16l1^{\Delta CD4}$ mice compared to control $Atg16l1^{fl/fl}$ littermates, whereas anti-CBir1 $IgG_{2b}$ and $IgG_{2c}$ levels were comparable (*Figure 3F*). Furthermore, CBir1-specific $IgG_1$ and IgA were already detectable in young $Atg16l1^{\Delta CD4}$ mice (data not shown). In

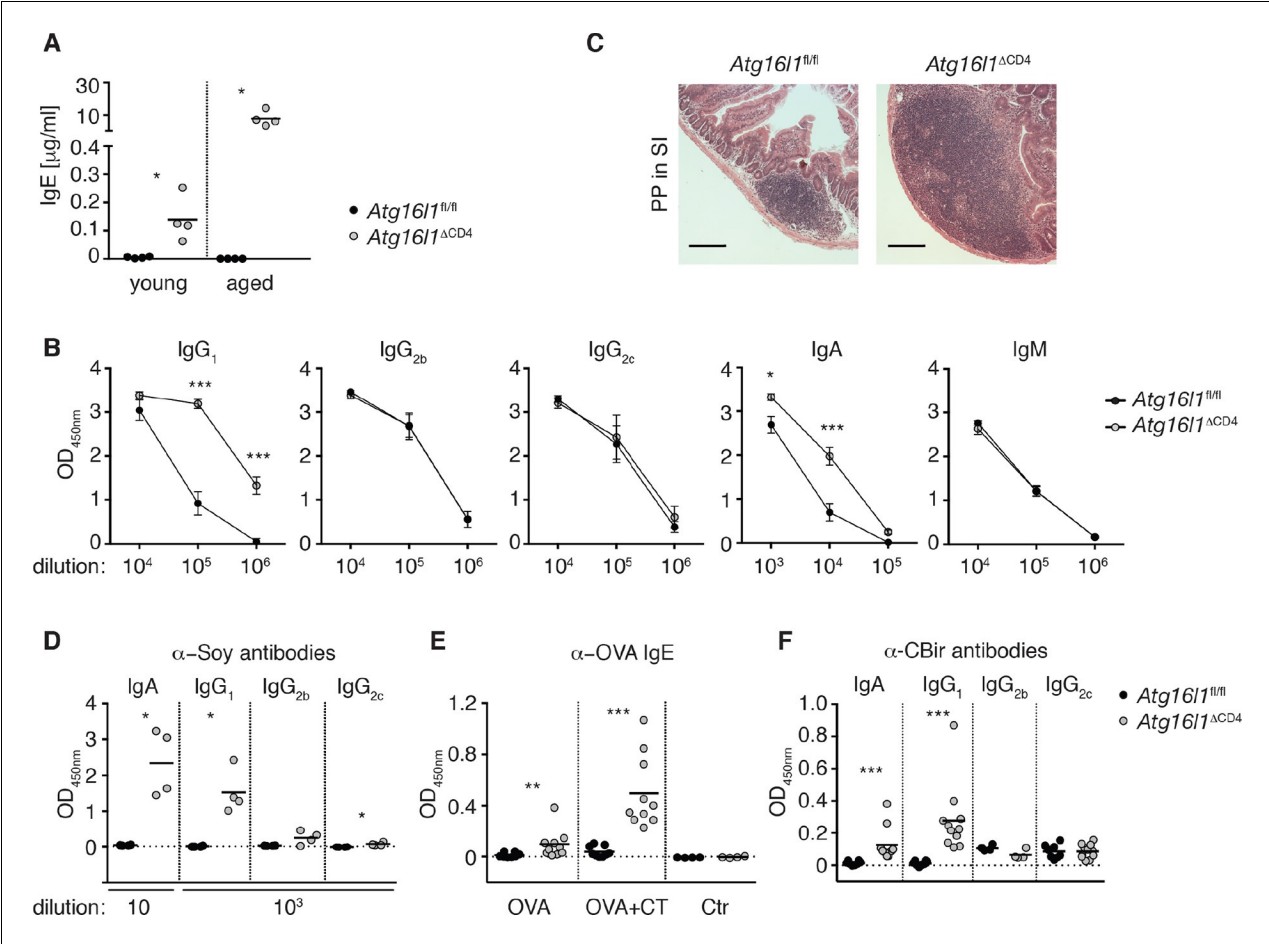

**Figure 3.** *Atg16l1*[ΔCD4] mice develop elevated $T_H2$-associated antibodies against intestinal luminal antigens. (A) Serum IgE concentrations in cohorts of young and aged *Atg16l1*[ΔCD4] and *Atg16l1*[fl/fl] littermates were measured by ELISA. (B) Serum antibody IgG$_1$, IgG$_{2b}$, IgG$_{2c}$, IgA and IgM isotype levels in aged *Atg16l1*[ΔCD4] and *Atg16l1*[fl/fl] littermates were measured by ELISA. (C) Representative photomicrographs of H&E stained sections of Peyer's patch (PP) in the SI (jejunum) of aged *Atg16l1*[ΔCD4] and *Atg16l1*[fl/fl] littermates, scale bar 150 μm. (D) Serum levels of Soy-specific IgA, IgG$_1$, IgG$_{2b}$, IgG$_{2c}$ antibodies in aged *Atg16l1*[ΔCD4] and *Atg16l1*[fl/fl] littermates were measured by ELISA. (E) Young *Atg16l1*[ΔCD4] and *Atg16l1*[fl/fl] littermates were fed with ovalbumin (OVA) alone or with cholera toxin (CT) as described in methods and levels of OVA-specific serum IgE were measured 8 weeks after first challenge by ELISA. (F) Levels of CBir1-specific IgA, IgG$_1$, IgG$_{2b}$ and IgG$_{2c}$ antibodies in serum of aged *Atg16l1*[ΔCD4] and *Atg16l1*[fl/fl] littermates were measured by ELISA, serum was diluted 50x. Data are representative from at least two independent experiments with at least three mice per group (A-D) or combined from two (E) or three (F) independent experiments with at least three mice per group. Each dot represents an individual mouse and horizontal bars denote means (A,D,E,F). Serum isotype levels are shown as mean ± s.e.m (B). Statistical significance was determined using the Mann–Whitney test (A,D-F) or two-way analysis of variance (ANOVA) with Bonferroni's correction for multiple comparisons (B), *p<0.05; **p<0.01; ***p<0.001. SI – small intestine. Young mice: 8–12 weeks old, aged mice > 5 months old.

The following figure supplement is available for figure 3:

**Figure supplement 1.** Dysregulated humoral responses in young *Atg16l1*[ΔCD4] mice.

contrast, increased $T_H2$ cell-associated antibody responses were not mounted in young *Atg16l1*[ΔCD4] mice following oral infection either with the Gram-negative bacterium *Helicobacter hepaticus* or with the nematode parasite *Trichuris muris* (*Figure 3—figure supplement 1D,E*). Taken together, these results indicate that the abnormal $T_H2$-associated antibody responses observed in *Atg16l1*[ΔCD4] mice preceded the development of overt inflammation and were selectively induced towards commensal microbiota and dietary antigens.

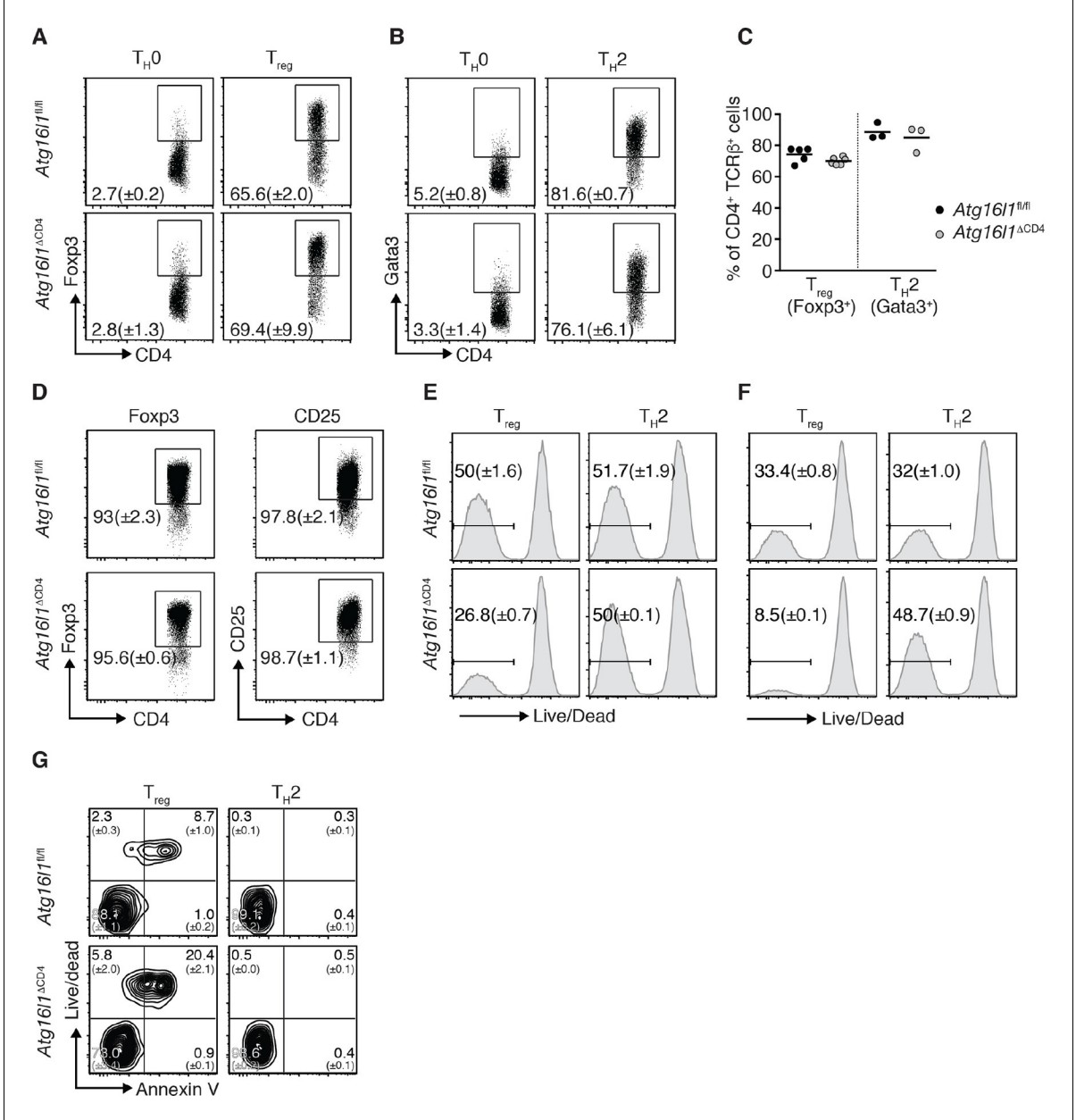

**Figure 4.** *Atg16l1* promotes survival of T_reg cells and limits T_H2 cell survival. (A,B) *Atg16l1*^ΔCD4 or *Atg16l1*^fl/fl naïve CD4+ T cells were cultured in T_H0, T_reg, or T_H2 polarizing conditions for 48 hr and analyzed by FACS. Representative FACS plots show (A) Foxp3 and (B) Gata3 expression (gated on CD4+ TCRβ+ T cells). (C) Frequencies of T_reg cells (Foxp3+) and T_H2 cells (Gata3+) arising from *Atg16l1*^ΔCD4 or *Atg16l1*^fl/fl naïve CD4+ T cells cultured in T_reg or T_H2 polarizing conditions for 5 days. (D) *Atg16l1*^ΔCD4 or *Atg16l1*^fl/fl T_reg cells were cultured with anti-CD3 (3 μg/ml) and anti-CD28 (1 μg/ml) for 48 hr, then maintained in the presence of IL-4 and IL-13 for a further 5 days before FACS analysis of Foxp3 and CD25 expression of live CD4+ T cells. (E,F) Naïve *Atg16l1*^ΔCD4 or *Atg16l1*^fl/fl CD4+ T cells were cultured with (E) 1 μg/ml or (F) 5 μg/ml anti-CD3 plus anti-CD28 (1 μg/ml) for 48 hr in T_reg or T_H2 polarizing conditions, then maintained in polarizing conditions for a further 5 days before FACS analysis of cell survival. Histograms show gates and frequencies of live CD4+ T cells. (G) Representative FACS plots of viability dye and Annexin V staining of T_reg cells and T_H2 cells from the cLP of young *Atg16l1*^ΔCD4 and *Atg16l1*^fl/fl littermates, gated on CD4+ TCRβ+ Foxp3+ (left panel), or CD4+ TCRβ+ Gata3+ (right panel). Data are representative from two (D,G) or three independent experiments (A,B,E,F), or are combined from three independent experiments (C). Each dot represents an individual cell culture (C) or data are shown as mean ± s.e.m (A,B,D-F). Numbers indicate percentage of cells in quadrants (G). cLP – colonic lamina propria. Young mice: 8–12 weeks old.

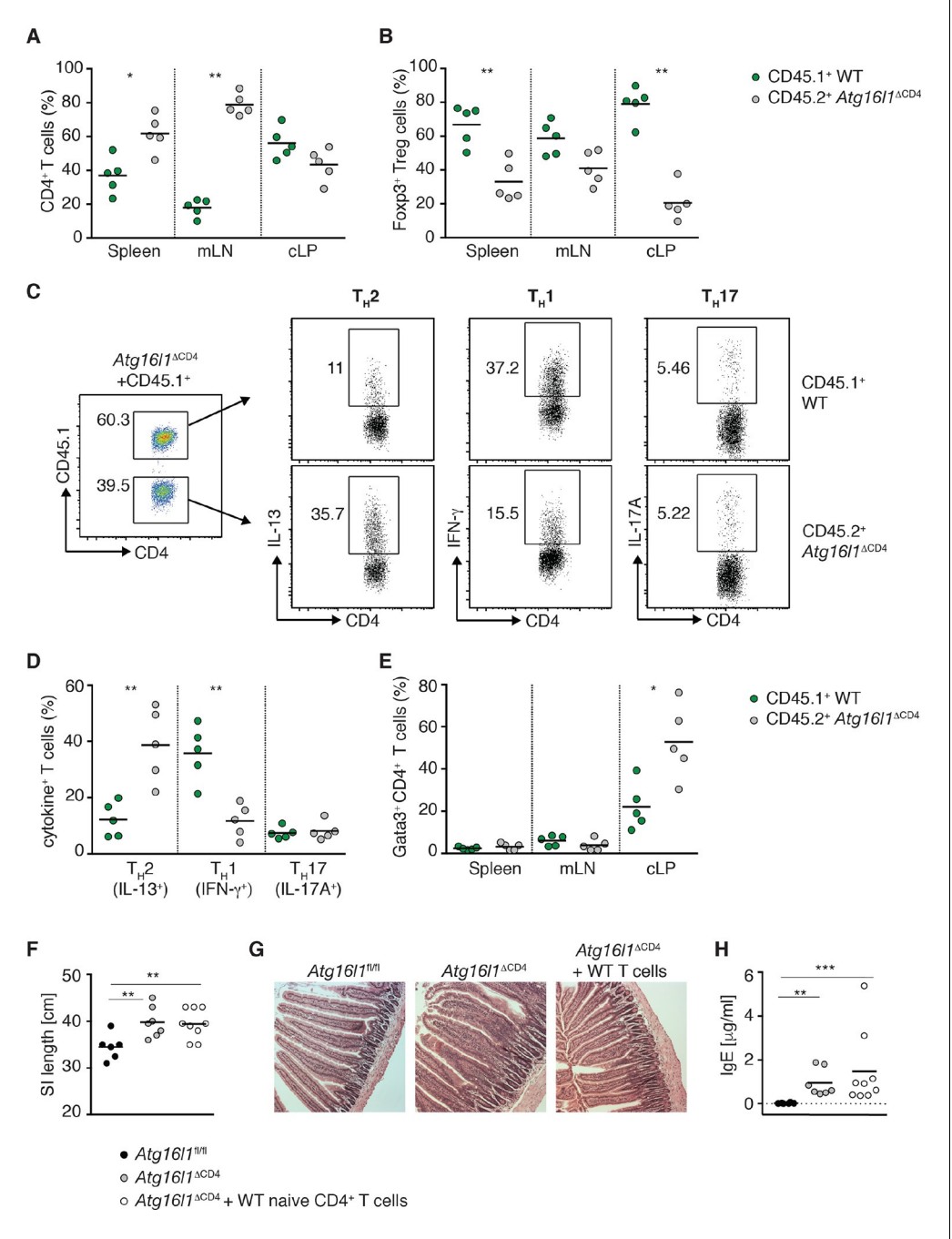

**Figure 5.** Autophagy contributes to the elevated T$_H$2 responses in *Atg16l1*$^{\Delta CD4}$ mice in a cell-intrinsic manner. Young *Atg16l1*$^{\Delta CD4}$ mice (CD45.2$^+$) were adoptively transferred with 4-5x10$^6$ naïve WT CD4$^+$ T cells (CD45.1$^+$) and analyzed 3 months later. (**A**) Frequencies of WT (CD45.1$^+$) and *Atg16l1*-deficient (CD45.2$^+$) CD4$^+$ T cells in the spleen, mLN and cLP. (**B**) Frequencies of WT (CD45.1$^+$) and *Atg16l1*-deficient (CD45.2$^+$) Foxp3$^+$ T$_{reg}$ cells in the spleen, mLN and cLP (gated on CD4$^+$ TCRβ$^+$ T cells). (**C**) Representative FACS plots showing gating of WT (CD45.1$^+$) and *Atg16l1*-deficient (CD45.1$^-$) CD4$^+$ T cells and expression of IL-13 (T$_H$2), IFN-γ (T$_H$1) and IL-17A (T$_H$17) in the cLP (gated on CD4$^+$ TCRβ$^+$ Foxp3$^-$ T cells). (**D**) Frequencies of WT (CD45.1$^+$) and *Atg16l1*-deficient (CD45.2$^+$) T$_H$2 (IL-13$^+$), T$_H$1 (IFN- γ$^+$) and T$_H$17 (IL-17A$^+$) cells among CD4$^+$ TCRβ$^+$ Foxp3$^-$ T cells in the cLP. (**E**) Frequencies of WT (CD45.1$^+$) and *Atg16l1*-deficient (CD45.2$^+$) Gata3$^+$ CD4$^+$ T cells in the spleen, mLN and cLP (gated on CD4$^+$ TCRβ$^+$ Foxp3$^-$ T cells). (**F**) SI lengths and (**G**) representative photomicrographs of jejunum of control untreated *Atg16l1*$^{fl/fl}$ or *Atg16l1*$^{\Delta CD4}$ littermates and reconstituted *Atg16l1*$^{\Delta CD4}$ mice, scale bar 150 μm. (**H**) Serum IgE concentrations in control untreated *Atg16l1*$^{fl/fl}$ or *Atg16l1*$^{\Delta CD4}$ littermates and adoptively transferred

*Figure 5 continued on next page*

*Figure 5 continued*

*Atg16l1*$^{\Delta CD4}$ mice were measured by ELISA. Data are representative of two independent experiments with at least four mice per group (**A-E,G**) or combined from two independent experiments (**F,H**). Each dot represents cells coming from the donor or the hosts within the individual transferred mouse (**A,B,D,E**) or each dot represents an individual mouse (**F,H**), horizontal bars denote mean. Numbers indicate percentage of cells in gates. Statistical significance was determined using the Mann–Whitney test, *p<0.05; **p<0.01. mLN - mesenteric lymph nodes, cLP – colonic lamina propria. Young mice: 10–12 weeks old.

The following figure supplement is available for figure 5:

**Figure supplement 1.** Reconstitution of intestinal CD4$^+$ T cell compartments in adoptively transferred *Atg16l1*$^{\Delta CD4}$ mice.

## Atg16l1 differentially regulates survival of T$_H$2 and T$_{reg}$ cells

Given apparent opposing effects of *Atg16l1* deficiency on T$_H$2 and T$_{reg}$ cells, we questioned whether the disruption of autophagy pathway affects the differentiation of these T cell subsets. We found that, under T$_H$2 or T$_{reg}$ polarizing conditions, differentiation of naïve CD4$^+$ T cells isolated from *Atg16l1*$^{\Delta CD4}$ or *Atg16l1*$^{fl/fl}$ littermates toward the Gata3$^+$ T$_H$2 or Foxp3$^+$ T$_{reg}$ cell phenotype was comparable (***Figure 4A–C***). As T$_H$2 cytokines can negatively affect T$_{reg}$ differentiation and stability (***Dardalhon et al., 2008***; ***Feng et al., 2014***), it was possible that outgrowth of T$_H$2 cells may also have contributed to the loss of intestinal T$_{reg}$ in *Atg16l1*$^{\Delta CD4}$ mice. We therefore isolated Foxp3$^+$ T$_{reg}$ cells from *Atg16l1*$^{\Delta CD4}$ and *Atg16l1*$^{fl/fl}$ littermates and activated them in vitro in the presence of IL-4 and IL-13. However, we did not find any evidence of T$_{reg}$ instability, as expression of Foxp3 and CD25 remained equally high in *Atg16l1*-deficient and WT T$_{reg}$ cells (***Figure 4D***).

We therefore examined whether autophagy deficiency influenced the survival of T$_H$2 or Foxp3$^+$ T$_{reg}$ cells. Thus, naïve CD4$^+$ T cells isolated from *Atg16l1*$^{\Delta CD4}$ or *Atg16l1*$^{fl/fl}$ littermates were activated for 48 hr with anti-CD3 and anti-CD28 antibodies and then rested for 5 days. Cells were kept in T$_H$2 or T$_{reg}$ polarizing conditions throughout the experiment. Following activation with different concentrations of anti-CD3 antibody, *Atg16l1*-deficient T$_H$2 cells exhibited comparable or improved survival relative to WT T$_H$2 cells (***Figure 4E,F***). In contrast, there was a 50–75% decrease in survival of *Atg16l1*-deficient T$_{reg}$ cells when compared to *Atg16l1*-sufficient T$_{reg}$ cells activated under the same conditions (***Figure 4E,F***). To establish whether autophagy-deficient T$_{reg}$ and T$_H$2 cells exhibited similarly distinct survival profiles in vivo CD4$^+$ T cells isolated from cLP of *Atg16l1*$^{\Delta CD4}$ or *Atg16l1*$^{fl/fl}$ littermates were stained with a viability dye and Annexin V. We observed that an increased proportion of *Atg16l1*-deficient intestinal T$_{reg}$ cells were dead or dying compared to WT T$_{reg}$ cells (***Figure 4G***). In contrast, *Atg16l1*-deficiency had no negative effect on the viability of intestinal T$_H$2 cells, which was comparable to WT controls (***Figure 4G***). Together, these results indicate that *Atg16l1*-deficiency does not impair the differentiation or stability of T$_{reg}$ cells and does not promote differentiation towards the T$_H$2 lineage. However, autophagy differentially impacts on the survival of mucosal T$_{reg}$ and T$_H$2 cells.

## Autophagy regulates intestinal T$_H$2 responses in a cell-intrinsic manner

As pT$_{reg}$ cells are required to control T$_H$2 responses at mucosal sites (***Mucida et al., 2005***; ***Curotto de Lafaille et al., 2008***; ***Josefowicz et al., 2012***), we examined whether the enhanced T$_H$2 phenotype in *Atg16l1*$^{\Delta CD4}$ mice could be corrected by reconstitution of the intestinal Foxp3$^+$ T$_{reg}$ compartment. We restored the pTreg population in young *Atg16l1*$^{\Delta CD4}$ mice at the age of 10–12 weeks, before the onset of intestinal pathology, through adoptive transfer of congenic WT naïve CD45.1$^+$ CD4$^+$ T cells. Recipients were sacrificed 3 months later, when control *Atg16l1*$^{\Delta CD4}$ littermates had developed intestinal inflammation. We detected CD45.1$^+$ donor CD4$^+$ T cells in all adoptively transferred *Atg16l1*$^{\Delta CD4}$ mice, but the level of reconstitution varied by the organ examined. In reconstituted *Atg16l1*$^{\Delta CD4}$ mice, donor WT CD4$^+$ T cells accounted for 37 ± 5% of total CD4$^+$ T cells in spleen and 18 ± 2% in mLN, whereas in the cLP they represented 56 ± 4% (***Figure 5A***). Thus, autophagy-deficient CD4$^+$ T cells had a survival disadvantage when compared to WT CD4$^+$ T cells within the intestinal mucosa. Overall, adoptive transfer of WT naïve CD4$^+$ T cells restored the total

frequencies of CD4$^+$ T cells in the cLP to levels comparable to control *Atg16l1*$^{fl/fl}$ mice (*Figure 5—figure supplement 1A*).

When we examined Foxp3$^+$ T$_{reg}$ cells, the survival advantage conferred by autophagy was even more apparent, with around 50% of the donor WT naïve CD45.1$^+$ T cells developing into Foxp3$^+$ pT$_{reg}$ cells in spleen, mLN and cLP of *Atg16l1*$^{\Delta CD4}$ recipients. As a result, the majority of Foxp3$^+$ T$_{reg}$ cells were of WT donor origin (67 ± 5% in spleen, 59 ± 4% in mLN and 80 ± 5% in cLP) (*Figure 5B*). Thus, adoptive transfer of WT-naïve CD4$^+$ T cells resulted in efficient reconstitution of Foxp3$^+$ pT$_{reg}$ cells in *Atg16l1*$^{\Delta CD4}$ mice; the total frequencies and numbers of T$_{reg}$ cells within the cLP of transferred mice were comparable with control *Atg16l1*$^{fl/fl}$ mice (*Figure 5—figure supplement 1B*). As such, we could utilize this system to determine whether excessive T$_H$2 cell accumulation in *Atg16l1*$^{\Delta CD4}$ mice was due to impaired mucosal pT$_{reg}$ cells or to a cell-intrinsic effect of *Atg16l1*-deficiency in T$_H$2 cells.

When we analyzed the frequencies of T$_H$2 cells in the cLP of reconstituted *Atg16l1*$^{\Delta CD4}$ mice, we observed significantly higher frequencies of Gata3$^+$ IL-13$^+$ T$_H$2 cells among *Atg16l1*-deficient CD4$^+$ CD4$^+$ T cells compared with the WT donor CD45.1$^+$ CD4$^+$ T cells (*Figure 5C–E*). Indeed, frequencies of IL-13$^+$ *Atg16l1*-deficient CD4$^+$ T cells in the cLP of pT$_{reg}$-reconstituted mice were comparable to those found in untreated *Atg16l1*$^{\Delta CD4}$ littermates (*Figure 5C,D*). In contrast, there was no difference in T$_H$17 cell frequencies between *Atg16l1*-deficient CD45.2$^+$ and WT CD4$^+$ T cells, and there was a significant decrease in T$_H$1 frequencies among *Atg16l1*-deficient CD4$^+$ T cells (*Figure 5C,D*). In line with these observations, adoptively transferred *Atg16l1*$^{\Delta CD4}$ mice had comparable total frequencies and numbers of T$_H$2 cells as observed in untreated *Atg16l1*$^{\Delta CD4}$ mice (*Figure 5—figure supplement 1C*). Thus, provision of WT pT$_{reg}$ cells did not rescue the increased T$_H$2 phenotype of *Atg16l1*-deficient CD4$^+$ T cells, indicating that autophagy directly regulates T$_H$2 cells through a cell-intrinsic mechanism. Consistent with this finding, *Atg16l1*$^{\Delta CD4}$ mice reconstituted with WT pT$_{reg}$ cells still developed intestinal pathology and elevated serum IgE levels comparable to those present in untreated *Atg16l1*$^{\Delta CD4}$ littermates (*Figure 5F–H*).

## Autophagy is essential for T$_{reg}$ cell homeostasis and control of effector T cell responses in the gut

Given that *Atg16l1*-deficiency significantly reduced the number of intestinal T$_{reg}$ cells in *Atg16l1*$^{\Delta CD4}$ mice, we hypothesized that T$_{reg}$ cells may be particularly reliant on autophagy compared to other subsets of CD4$^+$ T cells. Indeed, in WT mice we found that levels of autophagy were significantly higher in Foxp3$^+$ T$_{reg}$ cells compared to Foxp3$^-$ CD4$^+$ T cells, both constitutively and after TCR activation (*Figure 6A,B*). Together with our observations of impaired survival of *Atg16l1*-deficient Foxp3$^+$ T$_{reg}$ cells (*Figure 4E–G*), this suggested an important cell-intrinsic role for autophagy in the maintenance of T$_{reg}$ cells. This hypothesis was further strengthened by analyses of mixed bone marrow (BM) chimeras where irradiated *Rag1*$^{-/-}$ mice were reconstituted with a 1:1 mixture of BM cells from *Atg16l1*$^{\Delta CD4}$ mice and congenic WT C57BL/6mice (*Figure 6—figure supplement 1A*). In this setting, the reconstitution of CD4$^+$ T cells was severely hampered in the absence of functional autophagy and this deficiency was most pronounced in the T$_{reg}$ compartment of the spleen and cLP (*Figure 6—figure supplement 1B–E*), confirming that *Atg16l1*-deficiency decreases the ability of Foxp3$^+$ T$_{reg}$ cells to compete with WT T$_{reg}$ cells in a cell-intrinsic manner.

To definitively assess the cell-intrinsic requirement for autophagy in Foxp3$^+$ T$_{reg}$ cells we crossed *Atg16l1*$^{fl/fl}$ mice with mice expressing a YFP-Cre from the *Foxp3* locus (*Foxp3*$^{Cre}$ mice) (*Rubtsov et al., 2008*), generating *Atg16l1*$^{fl/fl}$::*Foxp3*$^{Cre}$ mice (hereafter denoted as *Atg16l1*$^{\Delta Foxp3}$) in which *Atg16l1* is selectively ablated in Foxp3$^+$ T$_{reg}$ cells. These mice allowed us to analyze the consequences of a lack of autophagy in T$_{reg}$ cells in the context of autophagy-competent CD4$^+$ T effector cells. As expected, *Atg16l1*$^{\Delta Foxp3}$ mice showed a significant reduction of *Atg16l1* expression in Foxp3$^+$ T$_{reg}$ cells, but not in CD4$^+$ Foxp3$^-$ T cells (*Figure 6—figure supplement 2A*). Although *Atg16l1*$^{\Delta Foxp3}$ mice appeared normal in early life, at around 5 months of age they developed a severe spontaneous inflammatory disease characterized by progressive weight loss, splenomegaly, lymphadenopathy and leukocyte infiltration in multiple tissues (*Figure 6C–E*). The gastrointestinal tract was particularly affected in aged *Atg16l1*$^{\Delta Foxp3}$ mice, with marked inflammation in the SI and colon (*Figure 6E,F*). Intestinal inflammation in aged *Atg16l1*$^{\Delta Foxp3}$ mice was characterized by massive accumulation of activated CD4$^+$ T cells in the intestinal LP and mLN (*Figure 7A–D* and *Figure 7—figure supplement 1A*). The cLP infiltrate in aged *Atg16l1*$^{\Delta Foxp3}$ mice contained a mixed

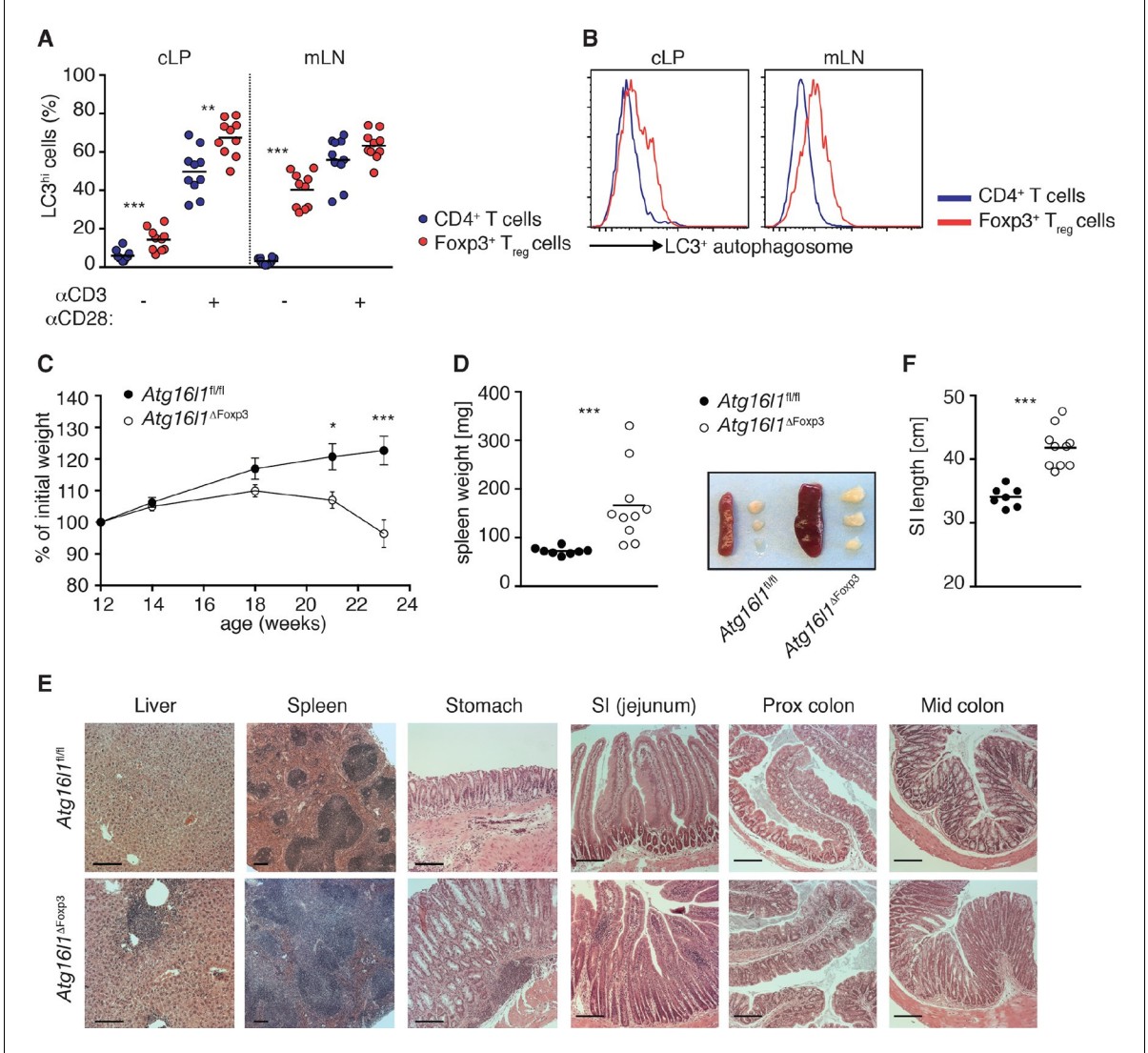

**Figure 6.** Aged *Atg16l1*^ΔFoxp3 mice develop spontaneous multi-organ inflammation. (A) LC3$^+$ autophagosome formation by Foxp3$^-$ CD4$^+$ T cells and Foxp3$^+$ T$_{reg}$ cells from cLP and mLN of WT mice in unstimulated cells or after overnight activation with α-CD3 (5 µg/ml) and α-CD28 (1 µg/ml). (B) Representative LC3 staining of unstimulated cells (gated on Foxp3$^+$ CD4$^+$ TCRβ$^+$ T$_{reg}$ cells or Foxp3$^-$ CD4$^+$ TCRβ$^+$ T cells). (C) Weight curves and (D) spleen weights and representative images of spleen and mLN of aged *Atg16l1*^ΔFoxp3 and *Atg16l1*^fl/fl littermates. (E) Representative photomicrographs of H&E stained sections of liver, spleen, stomach, SI (jejunum), proximal colon and mid-colon of aged *Atg16l1*^ΔFoxp3 and *Atg16l1*^fl/fl littermates, scale bar 150 µm. (F) Quantification of SI length. Data are combined from two to four independent experiments with two to five mice per group (A,D,F) or are representative of two to three independent experiments with two to five mice per group (B,C,E). Each dot represents an individual mouse and horizontal bars denote means (A,D,F). Data shown as mean ± s.e.m (C). Statistical significance was determined using two-way analysis of variance (ANOVA) with Bonferroni's correction for multiple comparisons (C) or using the Mann–Whitney test (A,D,F), *p<0.05; **p<0.01; ***p<0.001. mLN - mesenteric lymph nodes, SI – small intestine lamina propria, cLP – colonic lamina propria. Aged mice >5 months old.

The following figure supplements are available for figure 6:

**Figure supplement 1.** Impaired reconstitution of mixed bone marrow chimeras by *Atg16l1*-deficient T cells.

**Figure supplement 2.** Analysis of *Atg16l1* expression in *Atg16l1*^ΔFoxp3 mice.

---

population of T$_H$1, T$_H$17 and T$_H$2 effector cells, with a significant increase in the frequencies of IL-13$^+$ CD4$^+$ T$_H$2 cells (**Figure 7E,F**), although this T$_H$2 bias was not present in young *Atg16l1*^ΔFoxp3 mice (**Figure 7—figure supplement 1B**). In addition, we observed increased frequencies of Gata3$^+$ CD4$^+$

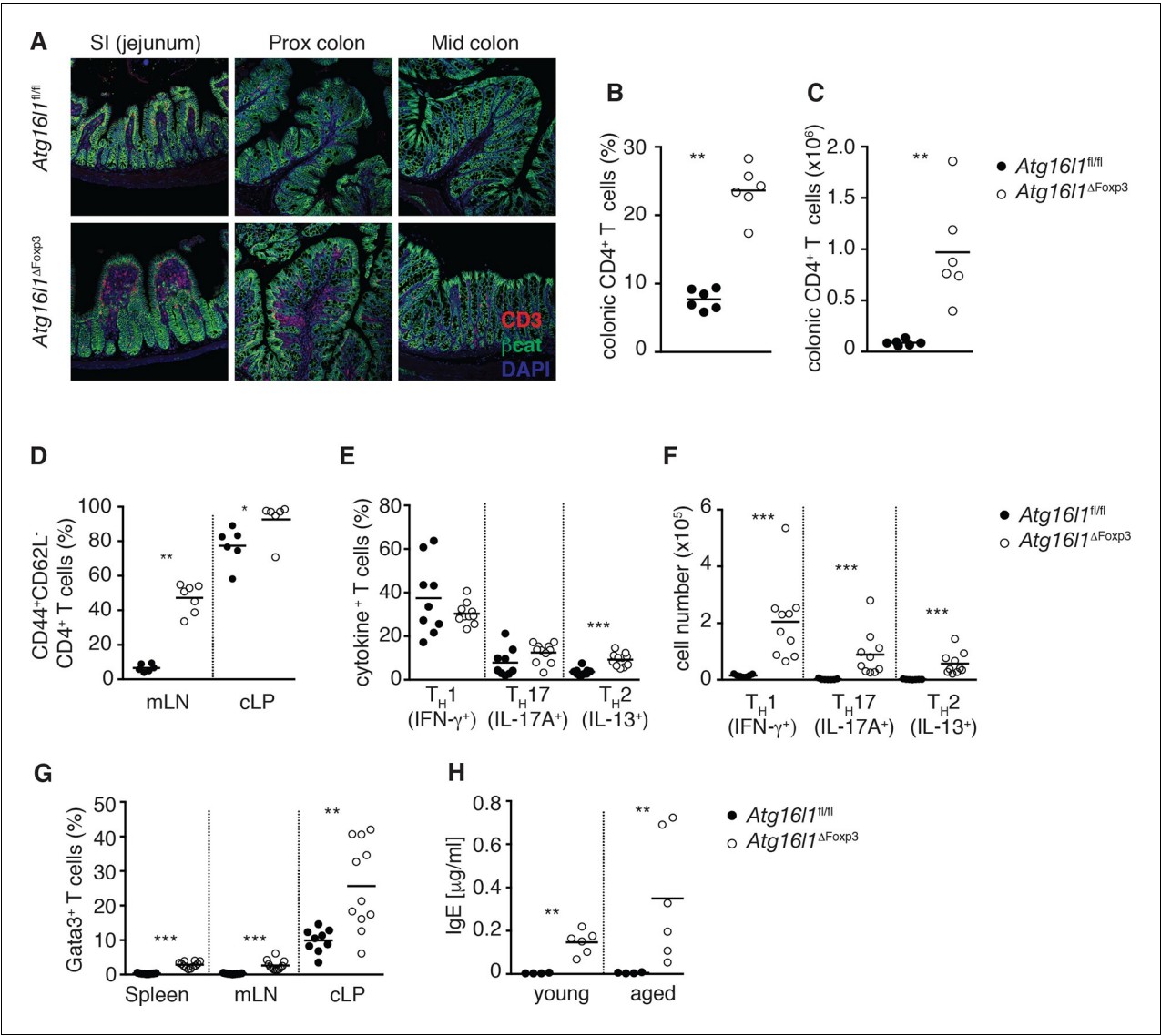

**Figure 7.** *Atg16l1*^ΔFoxp3 mice cannot control pro-inflammatory T_H effector responses. **(A)** Representative immunofluorescence images of small intestine and proximal and mid colon of aged *Atg16l1*^ΔFoxp3 and *Atg16l1*^fl/fl littermates stained for CD3 (red), β-catenin (green) and DAPI (blue). **(B)** Frequencies and **(C)** total numbers of cLP CD4^+ TCRβ^+ T cells in aged *Atg16l1*^ΔFoxp3 and *Atg16l1*^fl/fl littermates. **(D)** Frequencies of effector (CD44^+CD62L^-) CD4^+ T cells in the mLN and cLP of aged *Atg16l1*^ΔFoxp3 and *Atg16l1*^fl/fl littermates (gated on CD4^+ TCRβ^+ Foxp3^- T cells). **(E)** Frequencies and **(F)** total numbers of T_H1 (IFN-γ^+), T_H17 (IL-17A^+), T_H2 (IL-13^+) T cells in the cLP of aged *Atg16l1*^ΔFoxp3 and *Atg16l1*^fl/fl littermates (gated on CD4^+ TCRβ^+ Foxp3^- T cells). **(G)** Frequencies of Gata3^+ CD4^+ T cells in aged *Atg16l1*^ΔFoxp3 and *Atg16l1*^fl/fl littermates (gated on CD4^+ TCRβ^+ Foxp3^- T cells). **(H)** Serum IgE concentrations in *Atg16l1*^ΔFoxp3 and *Atg16l1*^fl/fl littermates were measured by ELISA. Data are combined from two to four independent experiments with two to five mice per group (**B-H**) or are representative of two independent experiments with two to five mice per group (**A**). Each dot represents an individual mouse and horizontal bars denote means. Statistical significance was determined using the Mann–Whitney test *p<0.05; **p<0.01; ***p<0.001. mLN - mesenteric lymph nodes, cLP – colonic lamina propria. Young mice: 8–12 weeks old, aged mice >5 months old.

The following figure supplement is available for figure 7:

**Figure supplement 1.** Additional characterization of *Atg16l1*^ΔFoxp3 mice.

T cells in the spleen, mLN and cLP of aged *Atg16l1*^ΔFoxp3 mice (**Figure 7G**). Analyses of humoral responses in aged *Atg16l1*^ΔFoxp3 mice revealed significantly elevated levels of circulating IgE and IgA, however IgG_1 levels were not increased (**Figure 7H** and **Figure 7—figure supplement 1C**). Thus, selective ablation of *Atg16l1* in Foxp3^+ T_reg cells led to intestinal inflammation that was characterized by accumulation of all T_H effector types, with a disproportionate increase in T_H2 responses in

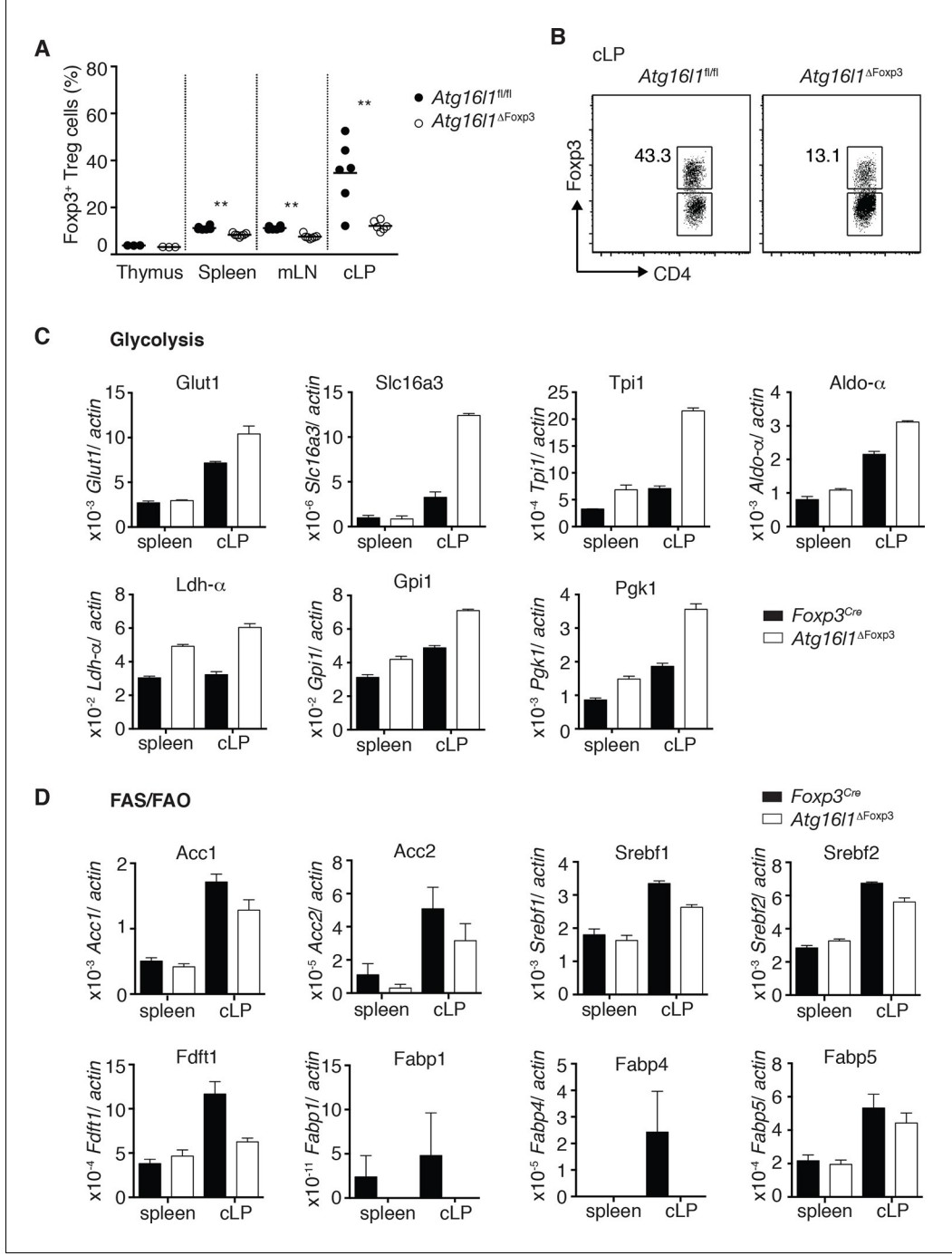

**Figure 8.** Cell-intrinsic autophagy is required for metabolic adaptation and survival of intestinal Foxp3[+] T_reg cells. (**A**) Foxp3[+] T_reg cell frequencies among CD4[+] TCRβ[+] T cells in $Atg16l1^{\Delta Foxp3}$ and $Atg16l1^{fl/fl}$ littermates and (**B**) representative FACS plots of Foxp3 expression in cLP CD4[+] T cells from young $Atg16l1^{\Delta Foxp3}$ and $Atg16l1^{fl/fl}$ littermates (gated on CD4[+] TCRβ[+] T cells). (**C**) qPCR analysis of glycolytic gene levels in sorted Foxp3[+] T_reg cells from spleen and cLP of young $Atg16l1^{\Delta Foxp3}$ and $Foxp3^{Cre}$ mice (sorted for CD4[+] TCRβ[+] YFP[+]). (**D**) qPCR analysis of FAS and FAO gene levels in Foxp3[+] T_reg cells from the spleen and cLP of young $Atg16l1^{\Delta Foxp3}$ and $Foxp3^{Cre}$ mice (sorted for CD4[+] TCRβ[+] YFP[+]). FAS: fatty acid synthesis, FAO: fatty acid oxidation, Glut1: glucose transporter 1, Slc16ac: solute carrier family 16 member 3 (lactic acid and pyruvate transporter), Tpi1: triosephosphate isomerase 1, Aldo–α: aldolase α, Ldh-α: lactate dehydrogenase α, Gpi1: Glucose phosphate isomerase 1, Pgk1: Phosphoglycerate kinase 1, Acc1: acetyl-CoA carboxylase 1, Acc2: acetyl-CoA carboxylase 2, Srebf1: sterol regulatory element binding transcription factor 1, Srebf2: sterol regulatory element binding transcription factor 2, Fdft1: farnesyl-diphosphate farnesyltransferase 1, Fabp: Fatty acid-binding protein. Data are representative from

*Figure 8 continued on next page*

*Figure 8 continued*

two (**C,D**) or three independent experiments (**A,B**). Each dot represents individual mouse (**A**) or data are shown as mean ± s.e.m (**C,D**). Gene expression levels are shown as mean ± s.e.m of three technical replicates (**C,D**). Numbers indicate percentage of cells in gates (**B**). cLP – colonic lamina propria. Young mice: 8–12 weeks old.

The following figure supplements are available for figure 8:

**Figure supplement 1.** *Atg16l1*-deficient colonic $T_{reg}$ cells exhibit increased cytokine secretion.

**Figure supplement 2.** Increased lipid uptake by intestinal $T_{reg}$ cells.

**Figure supplement 3.** $T_H2$ cells exhibit an enhanced glycolytic metabolic profile that is independent of autophagy.

aged mice. However, the breadth and magnitude of $T_H2$-associated responses were less pronounced in *Atg16l1*$^{\Delta Foxp3}$ mice compared to those observed in *Atg16l1*$^{\Delta CD4}$ mice.

When we examined the $T_{reg}$ cell compartment in *Atg16l1*$^{\Delta Foxp3}$ mice, we found significantly decreased frequencies of Foxp3$^+$ $T_{reg}$ cells in the spleen and mLN compared to *Atg16l1*$^{fl/fl}$ littermates, although thymic $T^{reg}$ cell frequencies were similar (*Figure 8A*). As found in *Atg16l1*$^{\Delta CD4}$ mice, intestinal LP Foxp3$^+$ $T_{reg}$ cells were severely depleted in *Atg16l1*$^{\Delta Foxp3}$ mice and those remaining exhibited significantly increased expression of effector $T_H$ cytokines (*Figure 8A,B* and *Figure 8—figure supplement 1A*). Thus, $T_{reg}$ cell-specific deletion of *Atg16l1* recapitulated the $T_{reg}$ cell deficits observed in *Atg16l1*$^{\Delta CD4}$ mice, showing that cell-intrinsic autophagy is essential for peripheral $T_{reg}$ cell homeostasis, especially in the intestine.

## Differential survival of autophagy-deficient $T_{reg}$ cells and $T_H2$ cells is associated with an altered metabolic profile

Finally, we investigated mechanisms that might underlie the striking survival defect of *Atg16l1*-deficient intestinal $T_{reg}$ cells. Analyses of key regulators of apoptosis revealed that *Atg16l1*-deficient $T_{reg}$ cells isolated from spleen and cLP had comparable expression of pro-apoptotic (*Bim, Bax*) and anti-apoptotic (*Bcl2*) genes as those isolated from control mice (*Figure 8—figure supplement 1B*). As recent evidence suggests that tissue-resident $T_{reg}$ cell populations may exhibit specialized metabolic adaptations (*Burzyn et al., 2013*), we compared the expression of metabolic genes by WT and *Atg16l1*-deficient $T_{reg}$ cells. Analyses of genes involved in glycolysis, fatty acid synthesis (FAS) and fatty acid oxidation (FAO), revealed that *Atg16l1*-deficient $T_{reg}$ cells had higher expression of glycolytic genes, including *Glut1, Slc16a3, Tpi1, Ldh-a, Aldo-a, Gpi1* and *Pgk1*, than control $T_{reg}$ cells (*Figure 8C*). Strikingly, this augmented glycolytic signature was much more pronounced in *Atg16l1*-deficient $T_{reg}$ cells isolated from cLP versus those from the spleen (*Figure 8C*). Conversely, expression of many key genes involved in FAS/FAO, including *Acc1, Acc2, Srebf1, Srebf2, Fdft1, Fabp1, Fabp4* and *Fabp5* was markedly decreased in *Atg16l1*-deficient $T_{reg}$ cells (*Figure 8D*). Again, these differences were most pronounced in the intestine; WT cLP $T_{reg}$ cells showed increased FAS/FAO gene expression compared to their spleen counterparts, whereas *Atg16l1*-deficient cLP $T_{reg}$ cells were not able to up-regulate the expression of FAS/FAO genes (*Figure 8D*). Thus, *Atg16l1*-deficiency profoundly influenced the expression of metabolic genes in intestinal $T_{reg}$ cells, with an altered balance of glycolytic and FAS/FAO gene expression. Further evidence of increased reliance on lipid metabolism by colonic $T_{reg}$ cells was provided by our observation that $T_{reg}$ cells isolated from the cLP showed markedly increased lipid uptake in comparison to mLN or spleen $T_{reg}$ cells (*Figure 8—figure supplement 2A,B*). A similar pattern was observed when we assayed expression of CD36, a fatty acid translocase that enhances FA uptake: colonic $T_{reg}$ cells showed increased expression of CD36 compared to splenic and mLN Treg cells (*Figure 8—figure supplement 2C,D*). Interestingly, we found that *Atg16l1*-deficient $T_{reg}$ cells showed comparable levels of lipid uptake and CD36 expression as their autophagy-sufficient counterparts (*Figure 8—figure supplement 2A–D*), suggesting that autophagy does not affect lipid uptake per se but rather affects lipid metabolism.

Together, these results demonstrate that cell-intrinsic autophagy is indispensable for Foxp3$^+$ $T_{reg}$ cell maintenance and function in peripheral tissues, particularly to suppress inflammatory responses

within the gastrointestinal tract. Decreased survival of *Atg16l1*-deficient T$_{reg}$ cells was associated with an altered metabolic profile, suggesting that autophagy plays an integral role in facilitating the metabolic adaptions required for long-term T$_{reg}$ cell survival in the intestine.

We next explored whether autophagy had a general effect on T cell metabolic profile and whether this might explain the differential effects on T$_H$2 cells and T$_{reg}$ cells. Evidence that this might be the case came from our observation that *Atg16l1*-deficient naïve CD4$^+$ T cells exhibited increased cell size compared with naïve CD4$^+$ T cells isolated from *Atg16l1*$^{fl/fl}$ littermates (*Figure 8—figure supplement 3A*). We therefore measured oxygen consumption rate (OCR), which is an indicator of oxidative phosphorylation (OXPHOS), and extracellular acidification rate (ECAR), an indirect indicator of aerobic glycolysis. We found that *Atg16l1*-deficient naïve CD4$^+$ T cells exhibited significantly increased OCR and ECAR, metabolic changes that are typically observed in activated CD4$^+$ T cells and are associated with increased aerobic glycolysis (*Figure 8—figure supplement 3B*).

As T$_H$2 cells have previously been reported to display an increased glycolytic rate compared to other T$_H$ subsets (*Michalek et al., 2011*; *Yang et al., 2013*), we hypothesized that they may be more resistant to the increased glycolysis that is induced in the absence of autophagy. As it was not possible to sort T$_H$2 cells from the cLP, we performed this analysis on in vitro cultures of T$_H$2 and T$_{reg}$ cells. We found that T$_H$2 cells were larger than T$_{reg}$ cells, expressed higher levels of c-Myc, a critical regulator of metabolic reprograming in activated T cells, and had markedly higher ECAR, all indicative of enhanced aerobic glycolysis (*Figure 8—figure supplement 3C–E*). Furthermore, while *Atg16l1*-deficient T$_{reg}$ cells showed higher expression of c-Myc, significantly increased levels of ECAR and OCR, and were larger than their control *Atg16l1*-sufficient counterparts, we observed constitutively high and comparable levels of glycolysis in *Atg16l1*-deficient and *Atg16l1*-sufficient T$_H$2 cells (*Figure 8—figure supplement 3C–E*). These patterns were recapitulated when expression of key metabolic genes were analyzed; T$_H$2 cells showed high expression of a panel of glycolytic genes irrespective of their autophagy *Atg16l1* genotype, whereas T$_{reg}$ cell expression of glycolytic genes was generally lower, unless the T$_{reg}$ cells were autophagy-deficient (*Figure 8—figure supplement 3F*). Taken together, these results suggest that the enhanced glycolytic metabolism constitutively employed by T$_H$2 cells makes them more resistant to the metabolic changes that occur in the absence of autophagy.

## Discussion

The unique challenges of the intestine necessitate complex mechanisms of tolerance and immune regulation to maintain homeostasis (*Izcue et al., 2009*). As altered mucosal CD4$^+$ T cell responses are implicated in intestinal diseases of increasing prevalence, including food allergies and IBD (*Maloy and Powrie, 2011*; *Berin and Sampson, 2013*), it is important to understand the factors that control effector and regulatory T cell homeostasis in the gut. Here, we identify Atg16l1 and autophagy as a new critical pathway regulating intestinal T$_{reg}$ and T$_H$2 responses.

Recent studies addressing the role of autophagy in distinct leukocyte populations have highlighted T cells as being very sensitive to perturbations in the autophagy pathway (*Ma et al., 2013*). Our data extend these findings by showing that autophagy is particularly important for the survival of CD4$^+$ T cells within the gut environment, as *Atg16l1* deletion in T cells led to a severe reduction of CD4$^+$ T cell numbers in the intestinal LP. This deficit was confirmed in mixed bone marrow chimeras, where *Atg16l1*-deficient CD4$^+$ T cells failed to reconstitute the intestinal LP compartment, and by the rapid outgrowth of adoptively transferred WT CD4$^+$ T cells in the colonic LP of *Atg16l1*$^{\Delta CD4}$ recipients. However, despite the reduction in intestinal CD4$^+$ T cells, *Atg16l1*$^{\Delta CD4}$ mice spontaneously developed progressive, chronic intestinal inflammation. To confirm their increased predisposition to develop intestinal pathology, we used an experimental model of IBD triggered by infection with *Helicobacter hepaticus* and concomitant treatment with anti-IL-10R mAbs (*Song-Zhao and Maloy, 2014*). This model induces severe typholocolitis that is T cell dependent and displays several features of human IBD pathology and does not require any specific genetic manipulation or chemical barrier disruption. We found increased intestinal pathology in *Atg16l1*$^{\Delta CD4}$ mice, confirming that *Atg16l1*-deficient T cells could mediate potent inflammatory responses in the gut. Thus, selective autophagy deficiency within T cells decreases the competitiveness of these cells and simultaneously predisposes to intestinal inflammation.

We found that $Atg16l1^{\Delta CD4}$ mice exhibited a drastic reduction in Foxp3$^+$ T$_{reg}$ populations in the cLP and SI LP, together with marked changes in intestinal T$_{reg}$ phenotype, including increased cell cycling and aberrant production of T$_H$ effector cytokines. The role of autophagy in Foxp3$^+$ T$_{reg}$ cells is not well defined. T cell-specific ablation of $Vps34$, which encodes a class III phosphatidylinositol 3-kinase that promotes autophagy, resulted in decreased frequencies of T$_{reg}$ cells in the thymus, spleen and lymph nodes (**Parekh et al., 2013**). However, as Vps34 also has autophagy-independent functions (**Backer, 2008**), it was unclear as to what extent these changes were due to impaired autophagy. Furthermore, we did not find any deficit in thymic T$_{reg}$ cell development in A$tg16l1$-deficient T cells. However, we observed that T$_{reg}$ cells isolated from the mLN and colonic LP had increased levels of autophagy compared to effector T cells, suggesting that autophagy is particularly important for the maintenance of T$_{reg}$ cells in the periphery. Indeed, we demonstrated that cell-intrinsic autophagy is indispensible for the maintenance and function of Foxp3$^+$ T$_{reg}$cells in the gastrointestinal tract, as selective deletion of $Atg16l1$ in the Foxp3$^+$ T$_{reg}$ compartment in $Atg16l1^{\Delta Foxp3}$ mice led to a loss of intestinal Foxp3$^+$ T$_{reg}$ cells and to severe inflammation of the small intestine and colon. In this context, it is pertinent that rapamycin, which induces autophagy through its inhibitory activity on mTOR, has been shown to promote expansion of T$_{reg}$ cells in vitro and in vivo (**Pollizzi and Powell, 2015**). Similarly, several small-molecule inducers of autophagy were shown to selectively promote the development of T$_{reg}$ cells in vitro (**Shaw et al., 2013**). Taken together with our findings, these observations suggest that boosting autophagy may represent a rational therapeutic approach to enhance T$_{reg}$ responses in the intestine.

How does autophagy intrinsically regulate T$_{reg}$ cell homeostasis? Our data indicate that autophagy is not required for the differentiation of Foxp3$^+$ T$_{reg}$ cells in vitro or in vivo for thymic generation of T$_{reg}$ cells in vivo. However, we found that $Atg16l1$-deficient T$_{reg}$ cells showed significantly decreased survival compared to WT T$_{reg}$ cells both in vitro and in vivo. As recent evidence indicates that T$_{reg}$ cells utilize a distinct metabolic program that favors lipid oxidation for energy provision (**MacIver et al., 2013**), one potential explanation is that autophagy regulates T$_{reg}$ cell metabolism and thereby their survival. Indeed, we found that $Atg16l1$-deficient T$_{reg}$ cells expressed a distinct metabolic profile to their WT counterparts, exhibiting increased expression of genes involved in glycolysis and reduced expression of genes involved in FAS/FAO. Fatty acid metabolism is emerging as a potent regulator of T cell responses and preferential utilization of FAO has been linked to T$_{reg}$ cell induction (**Lochner et al., 2015**). Although a recent report indicated that $de\ novo$ FAS was not required for Foxp3$^+$ T$_{reg}$ cell differentiation (**Berod et al., 2014**), optimal in vivo T$_{reg}$ cell function was associated with intrinsic lipid synthesis (**Zeng et al., 2013**). Furthermore, autophagy has been implicated in the regulation of fatty acid metabolism (**Singh et al., 2009**; **Lizaso et al., 2013**; **Kaur and Debnath, 2015**) and recent studies found that autophagy plays a key role in the generation of CD8$^+$ memory T cells (**Puleston et al., 2014**; **Xu et al., 2014**), which are heavily dependent on FAO for survival (**Pearce et al., 2009**; **O'Sullivan et al., 2014**). Thus, autophagy could play a similar survival role in T$_{reg}$ cells, by facilitating the degradation of intracellular lipid stores to release FAs that fuel FAO. Additionally, as degradation of intracellular lipids by autophagy is important to avoid lipotoxicity (**Galluzzi et al., 2014**), defective autophagy could lead to a toxic build up of intracellular lipids in intestinal T$_{reg}$ cells.

The imbalance between glycolysis and FAS/FAO observed in autophagy-deficient T$_{reg}$ cells could indicate that these cells have stalled in the activated/effector state and are unable to make the metabolic adaptations necessary for long-term survival. This is supported by our data showing that a higher proportion of autophagy-deficient T$_{reg}$ cells appear to be in cell cycle, but they have reduced expression of terminal differentiation markers. Consistent with our findings, a recent study reported that autophagy deficiency in T$_{reg}$ cells resulted in increased mTORC1 activation and glycolysis, leading to phenotypic instability, including expression of pro-inflammatory cytokines (**Wei et al., 2016**). However, the molecular mechanism behind decreased survival of autophagy-deficient T$_{reg}$ cells was not elucidated (**Wei et al., 2016**). It is striking that autophagy deficiency had a more detrimental effect on intestinal T$_{reg}$ cells than on those found in secondary lymphoid organs. Recent evidence suggests that tissue-resident T$_{reg}$ cells undergo tissue-specific adaptations, and metabolic changes are emerging as an important facet of such reprogramming (**Burzyn et al., 2013**; **Liston and Gray, 2014**). Taken together, our results suggest that autophagy endows intestinal T$_{reg}$ cells with the metabolic flexibility required to survive in the gut tissue, where essential growth factors may be in short supply (**Pearce et al., 2013**).

Paralleling decreased $T_{reg}$ responses in $Atg16l1^{\Delta CD4}$ mice, we observed a selective expansion of $T_H2$ cells in the intestinal LP that was already present in young mice and preceded the onset of overt pathology. Our subsequent analyses indicated that autophagy limits mucosal $T_H2$ cells through both cell-intrinsic and cell-extrinsic ($T_{reg}$-mediated) regulation. One possibility is that $Atg16l1$-deficient $T_H2$ cells may be somewhat resistant to $T_{reg}$ suppression. However, when we reconstituted $pT_{reg}$ cells in $Atg16l1^{\Delta CD4}$ mice we observed a negative correlation between the numbers of intestinal $T_{reg}$ cells and $T_H2$ cells (data not shown), suggesting that autophagy-deficient $T_H2$ cells are partially controlled by $T_{reg}$ cells. Our data strongly suggest that the intrinsic survival advantage of $Atg16l$-deficient $T_H2$ cells is primarily responsible for their outgrowth in the intestine. Indeed, we observed increased survival of $Atg16l1$-deficient $T_H2$ cells in vitro, suggesting that autophagy might directly inhibit $T_H2$ cell expansion. This concept is consistent with a previous study that reported enhanced survival of $T_H2$ cells in vitro when autophagy was inhibited and that autophagy mediated death of $T_H2$ cells during growth-factor withdrawal (*Li et al., 2006*). However, we provide evidence for an additional mechanism that could explain the preferential expansion of $Atg16l1$-deficient $T_H2$ cells in the intestine, related to the unique ability of $T_H2$ cells to cope with prolonged high levels of glycolysis. Our data contribute to accumulating evidence that a shift toward glycolysis is a general phenomenon observed when the autophagy pathway is perturbed in T cells. We observed characteristic signs of increased glycolysis in $Atg16l1$-deficient naïve CD4$^+$ T cells and $T_{reg}$ cells, such as increases in cell size, c-Myc levels and expression of glycolytic genes, as well as elevated ECAR. Others have reported a similar glycolytic shift in autophagy-deficient CD8$^+$ memory T cells (*Puleston et al., 2014*) and $T_{reg}$ cells (*Wei et al., 2016*). Interestingly, $T_H2$ cells have previously been shown to display an increased glycolytic rate compared to other $T_H$ subsets (*Michalek et al., 2011*; *Yang et al., 2013*). We confirmed the high levels of constitutive glycolysis in $T_H2$ cells and showed that these were comparable in $Atg16l1$-deficient and control $T_H2$ cells. Moreover, Gata3 activation was previously linked to induction of glycolysis after TCR activation in T cells, through induction of c-Myc, a critical regulator of metabolic reprograming (*Wang et al., 2011*; *Wang et al., 2013*; *Wan, 2014*). We therefore propose that in $T_H2$ cells Gata3 orchestrates metabolic adaptations that enable these cells to cope with prolonged high levels of glycolysis, thus making them resistant to metabolic changes enforced by autophagy deficiency. Overall, our results indicate that autophagy is a key pathway through which $T_H2$ responses are restrained in vivo. A lack of this restraint leads to a gradual loss of tolerance to intestinal antigens, as the excessive $T_H2$ responses in $Atg16l1^{\Delta CD4}$ mice led to production of IgG$_1$ and IgA antibodies toward commensal microbiota and dietary antigens that increased with age. Furthermore, $Atg16l1^{\Delta CD4}$ mice developed very high levels of circulating IgE, and mounted de novo IgE antibody responses toward introduced dietary antigen.

As polymorphisms in autophagy genes are linked to IBD susceptibility, our results point towards a novel mechanism that links impaired autophagy to intestinal inflammation through dysregulation of mucosal T cell responses. Previous studies focused on the role of *ATG16L1* and autophagy in myeloid cells and the intestinal epithelium. They suggested that impaired autophagy could result in reduced intestinal barrier integrity due to impaired Paneth cell function within the intestinal epithelial layer and elevated cytokine responses by macrophages and dendritic cells (*Cadwell et al., 2008*; *Saitoh et al., 2008*; *Lassen et al., 2014*). Our data add a further layer to the control of intestinal homeostasis by autophagy, by showing that autophagy impairment alters the local T cell compartment and promotes T cell driven intestinal pathology. We present compelling evidence that autophagy deficiency in $T_{reg}$ cells leads to a deficit in intestinal $T_{reg}$ cells and the development of severe intestinal pathology. Although the contribution of the $T_H2$ axis to IBD remains unclear (*Strober et al., 2002*; *Shale et al., 2013*), polymorphisms in *IL-4, IL-5* and *IL-13* have been implicated by GWAS in both CD and UC (*Van Limbergen et al., 2014*) and elevated levels of antibodies recognizing food and commensal antigens have been detected in IBD patients (*Lodes et al., 2004*; *Cai et al., 2014*). Moreover, as defective $T_{reg}$ and increased $T_H2$ responses at the mucosa are observed in food allergies and asthma, our findings might also have implications for these conditions. Indeed, epidemiological studies show an overlap between IBD and $T_H2$ driven diseases, such as atopic dermatitis and asthma (*Lees et al., 2011*). Furthermore, polymorphisms in the essential autophagy gene *Atg5* have recently been implicated in asthma susceptibility (*Martin et al., 2012*; *Poon et al., 2012*). Autophagy is an attractive therapeutic target and several autophagy modulating compounds are already in clinical trials for the treatment of various disorders (*Jiang and Mizushima, 2014*). Furthermore, natural dietary-derived compounds, including retinoid acid (*Isakson et al.,*

2010) and vitamin D (*Yuk et al., 2009*), have been shown to enhance autophagy. Taken together with our results, these findings raise the possibility that activation of autophagy through dietary or pharmacological modulation might have beneficial effects in disorders with a signature of decreased $T_{reg}$ and elevated $T_H2$ responses, including intestinal inflammation and various hypersensitivities.

## Materials and methods

### Mice

*Atg16l1*<sup>fl/fl</sup> mice were generated and provided by the H. Virgin laboratory (Washington University, Saint Louis, MO), as described (*Hwang et al., 2012*). *Atg16l1*<sup>fl/fl</sup> mice were crossed to B6.Cg-Tg (Cd4-cre)1Cwi/BfluJ (*CD4-Cre* mice) and B6.129(Cg)-Foxp3<sup>tm4(YFP/cre)Ayr</sup>/J (*Foxp3*<sup>Cre</sup> mice, Jackson Laboratory, Bar Harbor, ME) to generate *Atg16l1*<sup>ΔCD4</sup> and *Atg16l1*<sup>ΔFoxp3</sup> mice, respectively. All above strains, together with B6.SJL-CD45.1 (CD45.1<sup>+</sup>), B6 *Rag1*<sup>-/-</sup> (Jackson Laboratory), and B6 *Foxp3*<sup>hCD2</sup> mice (*Komatsu et al., 2009*) were bred and maintained under specific pathogen-free conditions. Unless stated otherwise, mice were analyzed at 8–12 weeks (young mice) or > 5 months of age (aged mice). In the gene expression analysis *Atg16l1*<sup>ΔFoxp3</sup> mice and *Foxp3*<sup>Cre</sup> mice were co-housed and age- and sex- matched. In all other experiments mice used were age- and sex-matched littermates that were kept co-housed throughout the experiments.

### T cell-mediated colitis

Experimental T cell-mediated colitis was induced by infection with *Helicobacter hepaticus* and concomitant IL-10R blockade as described (*Song-Zhao and Maloy, 2014*). Briefly, mice were infected with *H.hepaticus* (10<sup>8</sup> CFU per mouse) by oral gavage on three consecutive days and anti-IL-10R mAb (1B1.2) was administrated via i.p. injection (1 mg per mouse) on the first and seventh day of the infection. Mice were sacrificed 2 weeks after colitis induction.

### Histological assessment of intestinal inflammation

Mice were euthanized at indicated time points whereupon tissue sections were fixed in buffered 10% formalin and paraffin-embedded. Sections were then cut and stained with hematoxylin and eosin. Histological analysis of intestinal inflammation was performed as described (*Song-Zhao and Maloy, 2014*). Briefly, inflammation was graded semi-quantitatively on a scale from 0 to 3, for four criteria; (a) epithelial hyperplasia and goblet cell depletion, (b) lamina propria leukocyte infiltration, (c) area of tissue affected, and (d) markers of severe inflammation, including crypt abscesses, submucosal inflammation, and ulceration. Scores for individual criteria were totaled for an overall inflammation score between 0 and 12.

### Isolation of cells and flow cytometry analysis

Cell suspensions were prepared from the thymus, spleen, mLN, bone marrow and intestinal lamina propria as previously described (*Uhlig et al., 2006*). The following antibodies from eBioscience (Hatfield, UK) were used: anti-CD16/32 (93), anti-CD4 (GK1.5), anti-CD8α (53.6.7), anti-TCRβ (H57-597), anti-CD45 (30-F11), anti-CD44 (1M7), anti-CD62L (MEL-14), anti-CD45.1 (A20), anti-CD45.2 (104), anti-CD103 (2E7)), anti-CD69 (H1.2F3), anti-KLRG1 (2F1), anti-CD25 (7D4), anti-CD36 (No.72–1), anti-hCD2 (RPA-2.10), anti-CTLA4 (UC10-4B9), anti-GR.1 (RB6-8C5), anti-CD11b (M1/70), anti-Siglec F (E50-2440), anti-Gata3 (TWAJ), anti-Foxp3 (FJK-16s), anti-Ki67 (SolA15), anti-Helios (22F6), anti-Bcl2 (10C4), anti-PS6 (cupk43k), anti-IFN-γ (XMG1.2), anti-IL-17A (eBio17B7), anti-IL-13 (eBio13A). The following antibodies were from BioLegend (San Diego, CA): anti-CD138 (281–2), anti-CD161 (PK136), anti-F4/80 (BMB), anti-CD11b (M1/70). The following antibodies were from BD Biosciences (San Jose, CA): anti-B220 (RA3 6B2), anti-GL7 (GL7), anti-CD95 (Jo2), anti-CD3 (145-2C11), anti-CD19 (1D3), anti-Ly6C (AL-21), anti-Ly6G (1A8), anti-IgM (R6-60.2), anti-IgG1 (A85-1). Anti-c-Myc antibody was from Cell Signaling Technology (D84C12, Danvers, MA). Anti-Neuropilin1 polyclonal antibody was from R&D Systems (FAB566A, Minneapolis, MN). Fixable Viability Dye from eBioscience was used to stain dead cells. Annexin V staining was performed using eBioscience kit (88–08006) according to manufacture instructions. For intracellular cytokine staining cells were stimulated for 3h with PMA (100ng/ml) and Ionomycin (1 μg/ml) in the presence of Brefeldin A (10 μg/ml).

Autophagosome formation detection by flow cytometry was performed using FlowCellect Autophagy LC3 Antibody-based Assay Kit (FCCH100171, Merk-Millipore, Billerica, MA) according to the manufacturer's instructions and following cell surface markers staining. The Autophagy LC3 Antibody-based Assay Kit involves a permeabilization step to wash out cytosolic LC3-I, allowing for antibody-based detection of membrane bound LC3-II. For autophagy detection in WT $T_{reg}$ cells B6 Foxp3$^{hCD2}$ were used, as this allowed the detection of Foxp3$^+$ $T_{reg}$ cells on the basis of surface expression of hCD2 marker. All data were acquired using a Cyan ADP (Beckman Coulter, High Wycombe, UK) and analyzed using FlowJo software (Tree Star, Ashland, OR).

## CD4$^+$ T cell purification

Bulk CD4$^+$ T cells were purified from the spleen and mLN by negative selection as previously described (Coccia et al., 2012). Naïve CD4$^+$ T cells were then sorted as CD4$^+$ CD25$^-$ CD44$^-$ CD62L$^+$. $T_{reg}$ cells were sorted as CD4$^+$ CD25$^+$ when sorted from Atg16l1$^{\Delta CD4}$ and Atg16l1$^{fl/fl}$ mice and as CD4$^+$ YFP$^+$ when sorted from Atg16l1$^{\Delta Foxp3}$ and Foxp3$^{Cre}$ mice. Cells were sorted using an Astrios, Beckman Coulter MoFlo XDP or AriaIII BD Bioscience. Post-sort flow cytometry analyses confirmed that the purity of sorted populations was >97%.

## Adoptive transfer of naïve CD4$^+$ T cells

Naïve CD4$^+$ T cells from WT (CD45.1$^+$) mice were sorted as described above and transferred to Atg16l1$^{\Delta CD4}$ recipient (CD45.2$^+$) mice via intravenous injection (4-5x10$^6$ cells per mouse). Analysis of spleen, mLN and cLP CD4$^+$ T and $T_{reg}$ cells was performed 3 months after transfer.

## Generation of mixed bone marrow chimeras

BM cells were isolated from the tibia and femur of WT (CD45.1$^+$) mice and Atg16l1$^{fl/fl}$ or Atg16l1$^{\Delta CD4}$ (CD45.2$^+$) mice and injected i.v. at 1:1 ratio (a total of 1x10$^7$ cells per mouse) into lethally irradiated (1100 Rad, split dose) Rag1$^{-/-}$ recipients. Mice were allowed to reconstitute for at least 8 weeks before analysis.

## Immunization with ovalbumin (OVA)

For induction of OVA-specific IgE antibodies two treatment regimes were utilized. For OVA only immunization mice were fed three times by oral gavage with ovalbumin grade VII (5 mg per mouse, Sigma-Aldrich, St Louis, MO) with 21-day intervals between feeds. For adjuvanted immunization, mice were initially fed with OVA (5 mg per mouse) plus cholera toxin (10 µg per mouse, Biologial Compbell), after which they were fed twice with OVA only (5mg per mouse), with 21-day intervals between feeds.

## Infection with *Trichuris muris* and detection of *T. muris*-specific IgG$_1$

Mice were orally infected with ~200 *Trichuris muris* eggs. Serum was collected on day 34-post infection and assayed by ELISA for parasite-specific IgG$_1$. Ninety-six-well plates were coated with 5 µg/ml *T. muris* excretory/secretory antigen and incubated with serial two-fold diluted serum. Bound IgG$_1$ was detected using biotinylated anti-murine IgG$_1$ (AbD Serotec, Kidlington, UK).

## Lipid uptake measurement

Atg16l1$^{fl/fl}$ and Atg16l1$^{\Delta CD4}$ mice were injected i.p. with 50 µg of fluorescent 16-carbon fatty acid analog BODIPY C-16 (Molecular Probes) reconstituted in DMSO. Mice were culled 1 hr later and tissue collected for analysis by flow cytometry.

## Metabolic analysis using XF 96 extracellular flux analyzer

The real-time extracellular acidification rate (ECAR) and oxygen consumption rate (OCR) were measured using a XF 96 extracellular flux analyzer (Seahorse Bioscience, Billerica, MA). Briefly, naïve (CD62L$^+$CD44$^-$) CD4$^+$ T cells, or in vitro polarized $T_H$2 and $T_{reg}$ cells, were washed twice in assay medium (RPMI 1640 without sodium bicarbonate, 20 mM glucose, 1% FCS, 2mM pyruvate) and seeded at 3–4 x 10$^5$ cells per well in assay medium in a 96-well XF plate coated with poly-L-lysine (Sigma). T cells were rested for 1 hr at 37°C without $CO_2$ before analysis.

## Polarization and stimulation of CD4+ T cell subsets

Naïve CD4+ T cells were cultured ($3 \times 10^5$ cells/well) in 96-well plates coated with anti-CD3 mAb (5 µg/ml) and soluble anti-CD28 mAb (1 µg/ml) and kept in presence of IL-2 (100 U/ml). For $T_H0$ conditions anti-IL-4 (10 µg/ml) and anti-IFN-γ (10 µg/ml) mAb were added. Cultures were supplemented with IL-12 (10 ng/ml) and anti-IL-4 mAb (10 µg/ml) for $T_H1$ polarization; with IL-4 (20 ng/ml), anti-IFN-γ (20 µg/ml) and anti-IL-12 (10 µg/ml) for $T_H2$ polarization; and with TGF-β1 (5 ng/ml), anti-IFN-γ, anti-IL-4 mAb and anti-IL-12 (all 10 µg/ml) for induced $T_{reg}$ polarization. Sorted $T_{reg}$ cells were activated for 48h with anti-CD3 mAb (5 µg/ml) and soluble anti-CD28 mAb (1 µg/ml) plus IL-2 (100 U/ml) and then cultured with IL-4 (10 ng/ml), IL-13 (10 ng/ml) and IL-2 (100 U/ml) for 5 days. All cytokines were from R&D Systems. Anti-CD3 (145-2C11), anti-CD28 (37.51), anti-IFN-γ (XMG1.2), anti-IL-12 (C17.8) and anti-IL-4 (11B11) mAb were from eBioscience. Cells were cultured in RPMI-1640 Medium, 10% fetal calf serum, 2 mM L-glutamine, 100 U/ml of Penicillin/Streptomycin, and 0.05 mM 2-mercaptoethanol.

## Measurement of serum antibodies and cytokines

All immunoglobulin isotypes except for IgE were measured by enzyme-linked immunosorbent assay (ELISA) using the SBA Clonotyping System (Southern Biotech, Birmingham, AL). IgE concentration was determined using an anti-mouse IgE ELISA (BioLegend), according to manufacturer's instructions. For the detection of soy-specific, CBir-specific and *Helicobacter*-specific antibodies ELISA was performed with plates coated with purified soy antigen (5 µg/ml), CBir peptide (10 µg/ml) and soluble *Helicobacter* antigen (sHel antigen, 10 µg/ml) respectively. sHel antigen was prepared as previously described (*Kullberg et al., 1998*). For the detection of OVA-specific IgE, a sandwich ELISA was performed with biotinylated-OVA used for detection. MCPT-1 concentrations were measured by ELISA (eBioscience).

## Immunofluorescence microscopy

Colonic and small intestine tissue samples were formalin-fixed, paraffin-embedded and sectioned as per histological analysis. Sections were deparaffinized, rehydrated, and subjected to sodium citrate-based antigen retrieval, then stained with mouse pAb anti-β-catenin (610153, BD Bioscience), rabbit pAb anti-CD3 (ab5690, Abcam, Cambridge, UK) and secondary goat antibodies conjugated to AlexaFluor488 or 555 (Life Technologies, Carlsbad, CA). Slides were mounted with DAPI-containing Vectashield (Vector Laboratories, Burlingame, CA). Images were acquired with an Olympus Fluoview FV1000 confocal microscope and Olympus Fluoview Software (Olympus, Tokyo, Japan).

## Western blotting analysis

CD4+ T cells purified by negative selection were lysed in RIPA buffer containing protease inhibitor cocktail (Roche, Basel, Switzerland). Protein levels were normalized by Biorad DC protein assay (Bio-Rad Laboratories, Hercules, CA), resolved by SDS-PAGE and, following transfer onto nitrocellulose membranes, were blotted with anti-LC3 antibody (L7543; Sigma-Aldrich) and anti-tubulin antibody (sc5286, Santa Cruz Biotechnology, Dallas, TX), and secondary HRP conjugated anti–rabbit antibody (7074S, Cell Signaling Technology).

## Fluidigm gene expression analysis

CD4+ T cells and $T_{reg}$ cells were sorted for each population based on surface marker and YFP expression from spleen and cLP of $Atg16l1^{\Delta Foxp3}$ and $Foxp3^{Cre}$ mice. Two hundred cells/population were sorted in triplicates from a total of four (spleen) or six (cLP) mice per group. Alternatively, 250 cells from in vitro polarized populations of $T_H2$ and $T_{reg}$ cells were sorted from triplicate culture wells. RNA was reverse transcribed and cDNA was pre-amplified using the CellsDirect OneStep q-RT kit (Invitrogen). The selected autophagy, apoptotic and metabolic genes were amplified and analyzed for expression using a dynamic 48x48 array (Biomark Fluidigm) as previously described (*Tehranchi et al., 2010*). Data were analyzed using the $2^{-\Delta Ct}$ method, and the results were normalized to actin or HPRTprt expression.

## Statistical analysis

For weight curves and antibody titers, p-values were determined by two-way ANOVA with Bonferroni post-tests. For the metabolic analysis using XF 96 extracellular flux analyzer, p-values were determined using unpaired Student's t-test. For all other experiments, p-values were determined by nonparametric Mann–Whitney test. Differences were considered statistically significant when $p < 0.05$ ($* < p0.05$, $**p < 0.01$, $***p < 0.001$). Data are shown as mean $\pm$ s.e.m. Statistics were calculated using GraphPad Prism 6 software. For in vivo experiments, sample size was determined by power analysis using power of trial software, which calculates a power value based on $X^2$ test statistics. Calculated required sample sizes were applied whenever possible. No mouse was excluded from the analysis. With the exception of histological assessment of intestinal inflammation, experimenters were not 'blinded' to allocation of animals to experimental groups.

## Acknowledgements

We thank Herbert W Virgin and Thaddeus S Stappenbeck for mice with *lox*P-flanked *Atg16l1* alleles, Naren Srinivasan for insight and comments on the manuscript and Duncan Howie, Holm Uhlig and members of the Kevin Maloy and Fiona Powrie groups for helpful discussions and technical help. The analysis of the B cell compartment was carried out using methods developed by the 3i consortium (www.immunophenotyping.org).

## Additional information

### Funding

| Funder | Grant reference number | Author |
|---|---|---|
| Wellcome Trust | Graduate Student Scholarship 097112 | Agnieszka Martyna Kabat |
| Wellcome Trust | 100290 | Simon P Forman<br>Richard K Grencis |
| Wellcome Trust | 100156 | Simon P Forman<br>Richard K Grencis |
| Medical Research Council | MR/K011898/1 | Johanna Pott<br>Kevin J Maloy |
| Wellcome Trust | 102972 | Kevin J Maloy |

The funders had no role in study design, data collection and interpretation, or the decision to submit the work for publication.

### Author contributions

AMK, OJH, TR, AEM, AL, LA-D, SPF, JP, Conception and design, Acquisition of data, Analysis and interpretation of data, Drafting or revising the article; CFP, Assisted in the in vivo experiments, Acquisition of data, Analysis and interpretation of data, Drafting or revising the article; RKG, QS, AKS, KJM, Conception and design, Analysis and interpretation of data, Drafting or revising the article

### Author ORCIDs

Kevin J Maloy, http://orcid.org/0000-0001-5795-7688

### Ethics

Animal experimentation: All procedures on mice in this study were conducted in accordance with the UK Scientific Procedures Act (1986) under a project license (PPL 30/2872) authorized by the UK Home Office Animal Procedures Committee and approved by the Sir William Dunn School of Pathology Local Ethical Review Committee.

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
