## [Decision Letter]

Thank you for submitting your work entitled "The IBD-associated autophagy gene *Atg16l1* differentially regulates T_reg_ and T_H_2 cells to control intestinal inflammation" for consideration by *eLife*. Your article has been reviewed by two peer reviewers, and the evaluation has been overseen by Ivan Dikic as the Senior Editor.

The following individuals involved in review of your submission have agreed to reveal their identity: Vojo Deretic and Janneke Samsom (peer reviewers).

The reviewers have discussed the reviews with one another and the Reviewing Editor has drafted this decision to help you prepare a revised submission.

Summary:

The manuscript describes the role for autophagy in orchestrating intestinal T-cell responses. The authors convincingly show that autophagy is required for maintenance of Foxp3^+^ T_reg_ in vivo while, on the contrary, autophagy inhibits expansion of T_H_2 cells. In consequence, defects in autophagy in the CD4 T-cell compartment in vivo cause T_H_2 pathology by the lack of immune regulation due to reduced numbers of Foxp3 T_reg_ and is aggravated by a direct boosting of T_H_2 responses. These findings are novel in the field of autophagy and intestinal homeostasis.

Overall, the manuscript is nicely structured, convincing and guides the reader very well.

Essential revisions:

1) From the results describing Figure 5 it does not become clear why reconstitution with WT CD4^+^ T cells does not partially rescue the phenotype in *Atg16l1*^ΔCD4^ mice. On the basis of Figure 5 one would expect that 70% of the WT CD4 are T_reg_. Why are these cells unable to suppress the expanding *Atg16l1* deficient T_H_2 cells? Are *Atg161* deficient T_H_2 cells not sensitive to suppression by Foxp3^+^ T_reg_? This should be investigated in more detail. For example, is this expansion independent of IL-2 and can the Foxp3 T_reg_ therefore not suppress as easily?

2) As noted in the Abstract the role of autophagy in limiting mucosal T_H_2 expansion was unexpected. Although the authors suggest in the Discussion that Gata3 may promote peripheral T-cell proliferation and maintenance they do not explain how they think this works mechanistically. Are T_H_2 cells independent of autophagy for their metabolism? Is it possible to study the metabolic function of in vitro differentiated *Atg16l1*^ΔCD4^ T_H_2 and control T_H_2 cells in a similar experiment as has been performed in Figure 8? How does autophagy inhibit? This should be better explained as expansion of T_H_2 in *Atg16l1*^ΔCD4^ is a major finding in the study.

3) The effects of autophagy on fatty acid metabolism seems to be the best available explanation for the T_H_2 and T_reg_ disregulation but the experiments are a bit correlative here. The intestinal-specific defect seems to argue against that, although this reviewer does agree with the elegant thinking and correlations by the authors.

[Editors' note: further revisions were requested prior to acceptance, as described below.]

Thank you for resubmitting your work entitled "The IBD-associated autophagy gene *Atg16l1* differentially regulates T_reg_ and T_H_2 cells to control intestinal inflammation" for further consideration at *eLife*. Your revised article has been favorably evaluated by Ivan Dikic (Senior editor and Reviewing editor). The manuscript has been improved but there are some remaining issues that need to be addressed before acceptance, as outlined below:

The title should provide a clear indication of the biological system under investigation and it should avoid specialist abbreviations and acronyms where possible.

It is recommended to delete IBD from the title if you agree and this does not affect the main message of your paper:

"The autophagy gene *Atg16l1* differentially regulates T_reg_ and T_H_2 cells to control intestinal inflammation"

---

## [Author Response]

*Essential revisions: 1) From the results describing Figure 5 it does not become clear why reconstitution with WT CD4^+^ T cells does not partially rescue the phenotype in Atg16l1* δ

*CD4 mice. On the basis of Figure 5 one would expect that 70% of the WT CD4 are T_reg_. Why are these cells unable to suppress the expanding Atg16l1 deficient T_H_2 cells? Are Atg161 deficient T_H_2 cells not sensitive to suppression by Foxp3^+^ T_reg_? This should be investigated in more detail. For example, is this expansion independent of IL-2 and can the Foxp3 T_reg_ therefore not suppress as easily?*

The presentation of the data in the Figure 5 did not make clear to the reader what are the total levels of different T cell subsets in transferred *Atg16l1*^ΔCD4^ mice. We have now provided this information (Figure 5—figure supplement 1, and in the subsection “Autophagy regulates intestinal T_H_2 responses in a cell-intrinsic manner“), which shows that the frequencies and numbers of T_reg_ cells in reconstituted *Atg16l1*^ΔCD4^ mice are equivalent to those found in control *Atg16l1*^fl/fl^ mice. Therefore, although 70% of the total T_reg_ cells in the adoptively transferred *Atg16l1*^ΔCD4^ mice are of WT origin, the total T_reg_ cell levels are comparable with control *Atg16l1*^fl/fl^ mice.

To address whether autophagy-deficient T_H_2 cells were resistant to T_reg_ cell suppression, we examined whether there was any correlation between the frequencies of *Atg16l1*-deficient-T_H_2 cells and T_reg_ cells in the cLP of adoptively transferred *Atg16l1*^ΔCD4^ mice. This revealed a negative correlation, i.e. mice with higher levels of intestinal T_reg_ cells had the lowest levels of *Atg16l1*-deficient-T_H_2 cells (data not shown), suggesting that autophagy-deficient T_H_2 cells may be restricted to some extent by WT T_reg_ cells. However, the key conclusion from the data presented in Figure 5 (together with the supplemental data described above), is that despite efficient reconstitution of the T_reg_ cell compartment, the adoptively transferred *Atg16l1*^ΔCD4^ mice still develop increased frequencies of autophagy-deficient T_H_2 cells. The simplest explanation for this result is that autophagy regulates T_H_2 cell accumulation in cell-intrinsic manner. This is supported by the results in Figure 4 showing that autophagy-deficient T_H_2 cells exhibit increased survival compared to WT T_H_2 cells. However, we cannot completely rule out the reviewer's suggestion that autophagy-deficient T_H_2 cells might be somewhat resistant to inhibition by T_reg_ cells in the cLP in vivo and we therefore discuss this possibility in the Discussion.

To investigate whether autophagy-deficient T_H_2 cells might proliferate in an IL-2-independent manner, we assessed expression of IL-2Ra (CD25) on T_H_2 cells in the cLP of *Atg16l1*^ΔCD4^ mice and control mice. As shown below, we found comparable levels of CD25 on *Atg16l1*-deficient and WT T_H_2 cells in the cLP, as well as on polarized WT and *Atg16l1*-deficient T_H_2 cells in vitro. Furthermore, IL-2-driven STAT5 activation has been shown to contribute to T_H_2 lineage commitment (Zhu et al., Immunity, 2003; Liao et al., Nat. Immunol., 2008). Therefore, we think it unlikely that *Atg16l1*-deficient T_H_2 cells would develop or expand in an IL-2-independent manner.

Author response image 1.Autophagy deficiency does not influence expression of IL-2 receptor on T_H_2 cells (**A**) Expression of IL-2Ra (CD25) by T_H_2cells in the cLP of young *Atg16l1*^ΔCD4^ and *Atg16l1*^fl/fl^ littermates (gated on CD4^+^ TCRβ^+^ Foxp3^-^ Gata3^+^ T cells), data are representative of two independent experiments.(**B**) Expression of CD25 was measured in naïve *Atg16l1*^ΔCD4^ or *Atg16l1*^fl/fl^ CD4^+^ T cells cultured with anti-CD3 (5μg/ml) and anti-CD28 (1μg/ml) in T_H_2 polarizing conditions for 2 or 5 days, data are from one experiment.**DOI:**
http://dx.doi.org/10.7554/eLife.12444.023

*2) As noted in the Abstract the role of autophagy in limiting mucosal T_H_2 expansion was unexpected. Although the authors suggest in the Discussion that Gata3 may promote peripheral T-cell proliferation and maintenance they do not explain how they think this works mechanistically. Are T_H_2 cells independent of autophagy for their metabolism? Is it possible to study the metabolic function of in vitro differentiated Atg16l1* δ *CD4 T_H_2 and control T_H_2 cells in a similar experiment as has been performed in Figure 8? How does autophagy inhibit? This should be better explained as expansion of T_H_2 in Atg16l1* δ

*CD4 is a major finding in the study.*

The reviewers bring up a very interesting point and we have now performed additional experiments to compare the metabolic status of *Atg16l1*-deficient and WT Th2 cells. Our data in Figure 8 show increased expression of glycolytic genes in *Atg16l1*-deficient T_reg_ cells and recent studies reported similar glycolytic changes in autophagy-deficient T cells (Puleston et al., 2014; Wei et al., 2016). Thus, a consistent theme emerges whereby autophagy perturbation results in increased glycolytic metabolism in T cells. We therefore investigated the hypothesis that this glycolytic shift in autophagy-deficient CD4^+^ T cells negatively impacts on their survival, with the exception of T_H_2 cells, as previous publications suggested that T_H_2 cells have higher levels of glycolysis (Michalek et al., 2011). As it was not possible to isolate viable intestinal T_H_2 cells to perform comparative analysis of metabolic gene expression (as we were able to do with the T_reg_ cells that expressed a YFP marker), we analysed in vitro polarised T_H_2 cells. We confirmed that T_H_2 cells were highly glycolytic and found that this was independent of autophagy (Figure 8—figure supplement 3, and the last two paragraphs of the Results section). Mechanistically, we suggest that this highly glycolytic state is orchestrated by Gata3, which may enhance glycolysis through direct effects on c-Myc (Wang, 2013) and we found high levels of c-Myc expression in T_H_2 cells (Figure 8—figure supplement 3). As discussed in the Discussion section, we believe that T_H_2 cells are uniquely able to cope with prolonged high levels of glycolysis, and this may explain why they have a clear survival advantage over other T cells in the context of autophagy-deficiency.

3) The effects of autophagy on fatty acid metabolism seems to be the best available explanation for the T_H_2 and T_reg_ disregulation but the experiments are a bit correlative here. The intestinal-specific defect seems to argue against that, although this reviewer does agree with the elegant thinking and correlations by the authors.

We would like to argue that intestinal-specific defect fits with our observations and hypothesis. We saw the most striking defect in T_reg_ populations in the colonic LP in both *Atg16l1*^ΔCD4^ and *Atg16l1*^ΔFoxp3^ mice (Figure 2) and this is the same site where we observed that control T_reg_ cells had increased expression of genes implicated in lipid metabolism (Figure 8). To further support this observation we used an in vivo assay to measure lipid uptake by intestinal and systemic T_reg_ cells. We also measured their expression of the fatty acid translocase CD36. As shown in Figure 8—figure supplement 2 (described in the subsection “Differential survival of autophagy-deficient T_reg_ cells and T_H_2 cells is associated with an altered metabolic profile”), we found increased uptake of the fluorescent C16 lipid analogue and increased CD36 expression by colonic LP T_reg_ cells when compared with T_reg_ cells from the mLN and spleen. This further suggests that colonic T_reg_ cells are more adapted for lipid metabolism. Interestingly, short chain fatty acids (SCFA) produced by commensal bacteria were shown to facilitate pT_reg_ induction and to increase proliferation of intestinal T_reg_ cells. Although this was attributed to regulation of Foxp3 expression ^8,9^, our results tempt us to speculate that some of the beneficial effects could be due to SCFA acting as a metabolic fuel for intestinal T_reg_ cells.

[Editors' note: further revisions were requested prior to acceptance, as described below.]

The manuscript has been improved but there are some remaining issues that need to be addressed before acceptance, as outlined below: The title should provide a clear indication of the biological system under investigation and it should avoid specialist abbreviations and acronyms where possible. It is recommended to delete IBD from the title if you agree and this does not affect the main message of your paper: "The autophagy gene Atg16l1 differentially regulates T_reg_ and T_H_2 cells to control intestinal inflammation"

I can confirm that we have now made this change to the title, which now reads,

“The autophagy gene *Atg16l1* differentially regulates T_reg_ and T_H_2 cells to control intestinal inflammation”.